# SALMONN-omni: A Standalone Speech LLM without Codec Injection for Full-duplex Conversation

**Wenyi Yu**[1,2*]   **Siyin Wang**[1,2*]   **Xiaoyu Yang**[3]   **Xianzhao Chen**[2]   **Xiaohai Tian**[2]
**Jun Zhang**[2]   **Guangzhi Sun**[3]   **Lu Lu**[2]   **Yuxuan Wang**[2]   **Chao Zhang**[1†]
[1]Tsinghua University   [2]ByteDance   [3]University of Cambridge
{ywy22,wangsiyi23}@mails.tsinghua.edu.cn   cz277@tsinghua.edu.cn

## Abstract

In order to enable fluid and natural human-machine speech interaction, existing full-duplex conversational systems often adopt modular architectures with auxiliary components such as voice activity detectors, interrupters, conversation state predictors, or multiple LLMs. These systems, however, suffer from error accumulation across modules and struggle with key challenges such as context-dependent barge-in and echo cancellation. Recent approaches, most notably Moshi, simplify the pipeline by injecting audio codecs into the token space of a single LLM. However, such methods still incur significant performance degradation when operating on the speech rather than text modality. In this paper, we introduce SALMONN-omni, the first single, standalone full-duplex speech LLM that operates without audio codecs in its token space. It features a novel dynamic thinking mechanism within the LLM backbone, enabling the model to learn when to transition between speaking and listening states. Experiments on widely used benchmarks for spoken question answering and open-domain dialogue show that SALMONN-omni achieves at least 30% relative performance improvement over existing open-source full-duplex models and performs highly competitively to half-duplex and turn-based systems, despite using substantially less training data. Moreover, SALMONN-omni demonstrates strong performance in complex conversational scenarios, including turn-taking, backchanneling, echo cancellation and context-dependent barge-in, with further improvements achieved through reinforcement learning. Some demo conversations between user and SALMONN-omni are provided in the following repository https://github.com/bytedance/SALMONN.

## 1   Introduction

Large language models (LLMs) [1, 2, 3] have revolutionized problem-solving and task execution through text-based interaction. However, speech, being a more natural and intuitive modality, offers a smoother and more human-centric interface for human-machine communication. As a result, there is increasing interest in extending text-based LLMs with speech-processing capabilities. These efforts span a range of tasks, including advanced speech and audio understanding [4, 5, 6], text-driven speech synthesis [7, 8, 9] and interactive spoken dialogue systems [10, 11, 12].

While turn-based speech LLMs [10, 12, 13, 14, 15] support conversational artificial intelligence (AI) in a half-duplex manner, human conversation is inherently full-duplex, requiring the ability to listen, think, and speak simultaneously. This dynamic is evident in natural conversational behaviours such as frequent turn-taking, backchanneling, overlapping speech and barge-ins, all of which contribute to more fluid and engaging interactions. These rich, interactive communication patterns have spurred

---

[*]These authors contributed equally to this work.
[†]Corresponding author.

39th Conference on Neural Information Processing Systems (NeurIPS 2025).

growing interest in full-duplex speech LLMs, which seek to improve the naturalness, responsiveness and overall quality of human-AI conversations.

The recent release of GPT-4o (GPT-4omni) has demonstrated low-latency, emotionally expressive speech capabilities, further increasing global interest in full-duplex speech technologies. Among open research efforts, models such as Moshi [11], SyncLLM [16] and OmniFlatten [17] inject discrete audio codec tokens into pre-trained LLM backbones and fine-tune them to support flexible speech input and output. However, these approaches typically require large-scale training data to bridge the modality gap between text and speech and to mitigate catastrophic forgetting of original LLM textual knowledge. As a result, the speech synthesizer must infer whether to produce speech or silence based solely on the codec stream. While these models avoid the need for annotations of conversational dynamics, they frequently struggle with accurate timing of dialogue transitions [18]. Other approaches [19, 20, 21, 22] explicitly predict transitions between *speaking* and *non-speaking* states and integrate streaming speech encoders and synthesizers to enable real-time interaction. However, these systems typically rely on complex mechanisms rather than a standalone LLM to handle full-duplex functionality, resulting in added system complexity and suboptimal context integration. Some methods, such as [21, 22], propose cascaded systems with external modules like voice activity detection (VAD) and dialogue controllers to manage conversational flow, which suffers from error propagation. Others, including VITA [19] and Freeze-Omni [20], follow a "model-as-a-server" paradigm, running two interdependent LLM processes, with one dedicated to listening and the other to speaking, thus introducing additional computational and memory overhead.

This paper introduces SALMONN-omni, a standalone speech interaction model based on a novel codec-free, full-duplex spoken dialogue framework. By seamlessly integrating a streaming speech encoder, a single LLM backbone without audio codec injection, and a streaming speech synthesizer, SALMONN-omni supports simultaneous speech input and output within a unified end-to-end architecture. To autonomously manage dialogue state transitions in full-duplex interactions, we propose a periodic synchronisation mechanism coupled with a novel "thinking"' strategy, endowing the model with the temporal awareness that enables coherent and responsive real-time speech communication.

Our key contributions are summarized as follows:

- We present SALMONN-omni, the first standalone speech LLM that enables full-duplex human-AI conversations without injecting audio codecs into the LLM backbone. Unlike previous models, SALMONN-omni directly integrates a streaming speech encoder, an LLM, and a speech synthesizer, requiring careful synchronisation to process input and output speech simultaneously. As a standalone model, the model must also decide when to start and stop speaking on its own. To achieve this, we train the LLM to generate special tokens that control dialogue timing, treating these state transitions just like generating normal text tokens, based on the idea that **"your LLM is secretly a full-duplex predictor."**

- Experiments on widely used spoken question answering (QA) and conversation benchmarks demonstrate the strong performance of SALMONN-omni. In full-duplex mode, SALMONN-omni sets a new state-of-the-art (SOTA), achieving at least 30% relative improvement over previous best results. In half-duplex evaluation, it delivers highly competitive performance compared to recent turn-based models, including those trained on substantially larger datasets – some with up to 13 million hours of audio [21].

- SALMONN-omni shows strong performance in predicting the timing of turn-taking, backchanneling, and context-dependent barge-ins in free-form conversations. Additionally, to the best of our knowledge, we are the first to incorporate reinforcement learning (RL) to further improve the modeling of dialogue dynamics.

## 2 Related Work

### 2.1 End-to-end speech (interaction) LLMs

Compared with traditional modular conversational AI systems [23, 24, 25], end-to-end speech interaction LLMs have attracted great attention for their ability to support fluent, expressive, and emotionally rich spoken interactions. Depending on whether the model can listen and speak simultaneously, a core characteristic of human communication, recent end-to-end conversational AI systems can be broadly categorized into two types: *half-duplex (turn-based)* and *full-duplex* speech LLMs.

Currently, most speech LLMs operate in a half-duplex manner [13, 14, 19, 26, 27], including models such as GLM-4-Voice [15], Qwen2.5-Omni [22], and Kimi-Audio [21]. While these models can engage in turn-based speech conversations, they lack an internal duplex strategy for modeling dialogue dynamics like turn-taking. Instead, they rely on external VAD modules to alternate between listening and speaking states. As a result, they struggle with key aspects of natural conversation (*e.g.*, barge-ins and backchanneling) which require the ability to listen and speak simultaneously.

Full-duplex speech LLMs [11, 16, 17, 20, 28, 29] can process streaming speech input and output simultaneously, while also determining when to speak and when to stop. One straightforward approach to building such models involves injecting audio codecs into the LLM vocabulary, adopted by Moshi [11], syncLLM [16] and OmniFlatten [17]. Despite its conceptual simplicity, this approach demands large-scale speech-text paired data to prevent catastrophic forgetting. Even with extensive training, they typically lag behind similarly sized text-only LLMs in knowledge and reasoning and suffer from the modality gap. An alternative strategy used by VITA [19], Freeze-Omni [20] and MinMo [28] is to connect the LLM backbone with speech encoder and synthesizer through embeddings, without significantly hurting the LLMs. However, they are not standalone full-duplex since one LLM instance can only listen or speak, and they require two separate LLM processes to manage simultaneous listening and speaking. In contrast, SALMONN-omni introduces a novel duplex strategy that enables a single LLM to perform standalone full-duplex speech interaction.

Table 1: Categorization of speech interaction LLMs by full-duplex support.

| Half-duplex (turn-based) | Full-duplex | |
| --- | --- | --- |
| | standalone | non-standalone |
| GLM-4-Voice[15], Qwen2.5-Omni[22] | dGSLM[30], Moshi[11] | Freeze-Omni[20] |
| miniCPM-o[26], Baichuan-Audio[27] | SALMONN-omni | VITA[19], MinMo[28] |

## 2.2 Reinforcement learning (RL) for speech LLMs

RL has long been applied to advance speech tasks, for example, to minimize word errors for automatic speech recognition (ASR) [31, 32, 33, 34], to optimize task-oriented spoken dialogue procedure [35, 36], and to enhance emotional expressiveness in speech synthesis [37, 38, 39]. As for speech LLMs, recent RL methods such as proximal policy optimisation [40] and direct preference optimisation (DPO) [41] have been adopted by speech understanding models like Qwen2-Audio [6] and Step-Audio [42] to align speech LLM behaviour with human preference. However, RL remains unexplored in the context of full-duplex speech LLMs. To the best of our knowledge, SALMONN-omni is the first work to apply RL to improve the modeling of dialogue dynamics for full-duplex speech LLMs.

## 3 Methodology

Four challenges must be solved when implementing a full-duplex speech LLM. To implement SALMONN-omni as a standalone full-duplex model without codec injection, a novel codec-free (no audio codecs in LLM vocabulary) full-duplex spoken dialogue framework is proposed.

**The model must support streaming speech input and output.** Without relying on codecs, SALMONN-omni achieves real-time speech interaction by integrating an LLM backbone with a streaming speech encoder and a streaming speech synthesizer through hidden embeddings.

**The model must process both environmental sounds and the assistant's speech simultaneously.** In full-duplex conversations, the speech LLM must generate speech responses as the *assistant stream* while concurrently processing all incoming sounds (background sounds, user speech, assistant echo and *etc.*) as the *environment stream*. SALMONN-omni achieves this by interleaving LLM text response tokens, environment stream embeddings, and assistant stream embeddings into a single sequence, enabling the LLM backbone to model them jointly in an autoregressive manner. Note that Moshi [11] also jointly models these two streams, but with different purposes and design choices.

**The model must incorporate a sense of "time" to align audio and text modalities.** SALMONN-omni employs a periodic synchronisation mechanism, processing a fixed duration of input speech while generating a matching duration of speech responses in each time block for smooth interactions.

**The model must handle natural conversation dynamics such as turn-taking and barge-ins.** This requires the model to decide when to start and stop speaking based on contextual understanding. SALMONN-omni learns these state transitions using a novel "thinking" strategy. Instead of adding a separate full-duplex predictor, we propose that your LLM is secretly a full-duplex predictor and train it to generate state transition tokens as part of its normal output.

Notably, the last three features are key distinctions between SALMONN-omni and models such as VITA, Freeze-Omni, and MinMo, whose LLM decoders cannot listen and speak simultaneously. This limitation also prevents these models from achieving full-duplex dialogue in a standalone manner. The structure of SALMONN-omni is illustrated in Fig. 1, with a detailed explanation of its key components provided in the following subsections.

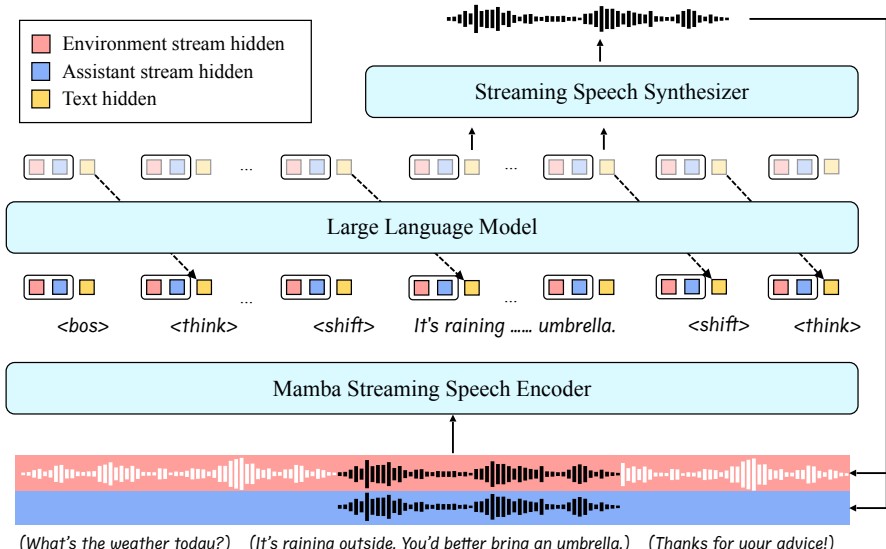

Figure 1: The architecture of SALMONN-omni. Two input streams, the environment stream and the assistant stream, are processed by the streaming speech encoder. Speech embeddings from both streams, along with textual embeddings, are fed into the LLM backbone in an interleaved manner. When in the speaking state, the streaming speech synthesizer takes the textual embeddings derived from the LLM backbone as input to produce speech responses.

## 3.1 Mamba streaming speech encoder

It is demonstrated that a streaming general-purpose encoder can be effectively trained using knowledge distillation from multiple pre-trained non-streaming encoders [43]. Following this approach, SALMONN-omni employs a streaming speech encoder designed to enhance generalisation in long conversational contexts, with Whisper-large-v3 [44] serving as the teacher model. Specifically, log-Mel filter bank features are extracted at a 100Hz frame rate, then downsampled to 50Hz using two convolutional layers. Next, two adjacent embeddings are concatenated into a single embedding, which is fed into a number of Mamba language model blocks [45]. An L1 loss function is applied to match the output features of the streaming encoder to those of the teacher model.

## 3.2 Streaming speech synthesizer

Our streaming speech synthesizer builds upon CosyVoice2 [46], adopting its fixed-length interleaved speech generation strategy. In this approach, input text and output speech tokens are interleaved so that for every $N$ text tokens, the model generates $M$ speech tokens in sequence. To support end-to-end integration with the LLM, we replace the original text input of the synthesizer with the LLM backbone's output embeddings. Several linear transformations is then applied to align these LLM embeddings with the input space expected by the streaming speech synthesizer.

### 3.3 Full-duplex strategy

**Interleaving multiple streams into a single stream.**    LLMs are traditionally trained to model a single sequence in an autoregressive manner. To enable pre-trained LLM backbones to effectively handle multiple concurrent streams in natural conversations, we propose interleaving these streams into a unified soft token sequence. Specifically, we divide a conversation into two primary streams: the assistant stream, representing the model's generated responses, and the environment stream, which includes all incoming audio, such as user speech, background noise, and echoes from the assistant's own speech. A full-duplex model must autoregressively generate the assistant stream while continuously attending to the environment stream. To facilitate this, we segment the conversation into fixed-duration time blocks. Within each block, SALMONN-omni first processes speech embeddings from the environment stream, then generates the corresponding assistant response. To further stabilise training and enhance performance, a dual-channel input mechanism is employed: in each time block, the model also receives speech embeddings from its own prior assistant outputs (including echo information) before generating the next textual response.

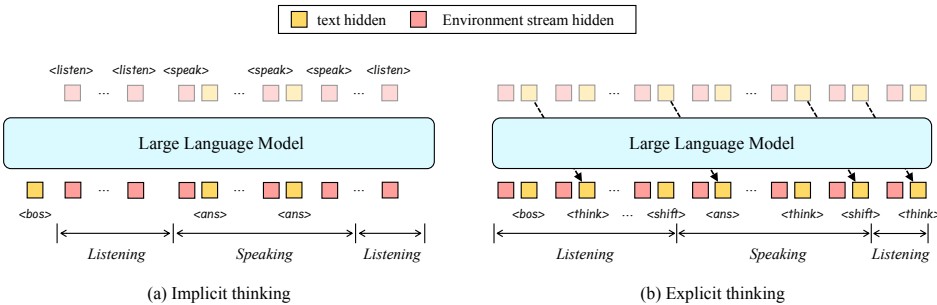

Figure 2: Illustration of "Implicit" and "Explicit" thinking strategies. The tokens on the top of the LLM are predicted by speech embeddings, while the bottom ones are predicted by textual embeddings and fed back to the input sequences to the LLM.

**"Thinking" strategy for handling dynamics in natural conversations.**    The difference between turn-based and full-duplex spoken dialogue models is the latter's ability to dynamically determine when to start and stop speaking. To enable this, we propose a novel "thinking" strategy that allows SALMONN-omni to learn the transitions between listening and speaking states. Within each time block, the model must decide whether a state transition should occur. We explored two variants of this strategy, which differ in whether the predicted state token from each time block is autoregressively fed as input to the next time block. An illustration of the two strategies is shown in Figure 2.

**Implicit "thinking".**    Under this strategy, the state determined by the model in each time block is not carried over in the LLM's input sequence. Whenever receiving an audio segment, the model determines the state as "listen" or "speak". If "listen", it continues to ingest the next audio time block. If "speak", it generates a specified number of textual tokens as the response before consuming the next audio segment. Notably, when a state transition from "listen" to "speak" occurs, a special token <shift> is sent to the input manually to initiate the text generation process.

**Explicit "thinking".**    This strategy does not distinguish between the prediction of state-related special tokens and the generation of the textual response. Consequently, a special token predicted by the model is autoregressively used as input for the subsequent time block. Two special tokens, <think> and <shift>, are introduced in this strategy. The special token <think> is utilized in two scenarios. First, when the model is in the listening state, it should generate <think> in each time block, which mimics the human thought process of deciding when to speak in a conversation. Second, due to the large difference in frame rates between text and speech modalities, the model generates <think> during the time blocks after completing the text response while waiting for a state transition from speaking to listening. The special token <shift> marks the state transitions of both "listening → speaking" and "speaking → listening".

The comparison of these two strategies is performed in Section 5.1, and the better performance achieved through explicit "thinking" demonstrates that training the LLM to generate a whole sequence

consisting of both normal responses and state-transition-related tokens naturally aligns with its intrinsic mechanism. In other words, **your LLM is secretly a full-duplex predictor**.

### 3.4 Training scheme

We propose a three-stage training procedure for SALMONN-omni, as illustrated in Fig 3.

**Stage 1: Connecting streaming encoder.** In stage 1, we focus on enabling the model with streaming speech understanding ability by connecting the Mamba streaming encoder with the LLM. The connector between the streaming encoder and the LLM is an MLP. Only the connector and LoRA on LLM are trained on ASR and QA tasks, with the encoder and LLM backbone frozen.

**Stage 2: Connecting streaming synthesizer.** In stage 2, SALMONN-omni is equipped with the speech generation capability and the whole model is trained in an end-to-end manner. In this stage, the streaming encoder and LLM backbone remain frozen. The connectors for the encoder and synthesizer, LoRA on LLM and the streaming synthesizer are trained on ASR, QA and multi-turn dialogue, including conversation dynamics such as barge-ins and backchanneling.

**Stage 3: RL for full-duplex modeling.** After the first two-stage training, SALMONN-omni demonstrates the capability to manage complex dynamics in free-form conversations, particularly in handling context-aware barge-ins and backchanneling. However, after the above supervised fine-tuning (SFT), the model exhibits a clear bias towards being interrupted and shows insufficient understanding of the dialogue semantics. Thus, we incorporate reinforcement learning to further enhance the full-duplex modeling ability. Concretely, DPO [41] is applied to barge-in and backchanneling tasks, while a subset of the SFT data from other tasks is retained to preserve overall performance.

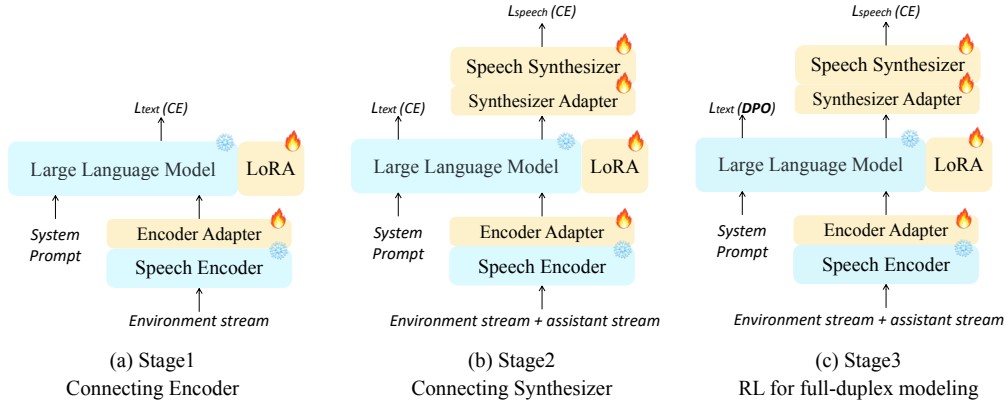

Figure 3: Three-stage training strategy for SALMONN-omni.

## 4 Experimental Setups

### 4.1 Model specifications

The Mamba streaming encoder in SALMONN-omni consists of 32 Mamba [45] LM blocks with a 2048-dimensional hidden state. The encoder generates embeddings at a frame rate of 25Hz, requiring downsampling of Whisper-large-v3 embeddings from 50Hz to 25Hz. This is achieved by concatenating every two adjacent embeddings into a single embedding. To ensure dimensional compatibility before applying L1 loss, we introduce a linear layer to align the embedding dimensions of the two models. We use Llama-3-8B-Instruct as the LLM backbone and use LoRA [47] with a rank of 32 and a scaling factor of 1.0 when finetuning the LLM backbone. Our streaming speech synthesizer is finetuned based on CosyVoice2-0.5B [46].

We set 80 milliseconds (ms) as the time block size and the model generates one textual token after listening to 80 ms of audio. Once the LLM backbone generates 4 tokens, 12 speech tokens will be generated by the streaming speech synthesizer, namely 480 ms speech. Thus, a delay of 320 ms

occurs between the moment the model initiates speech and the onset of audio output. The current design strikes an effective balance between response latency and speech generation quality.

## 4.2 Training tasks

SALMONN-omni is trained on ASR, spoken QA and multi-round conversation tasks. For ASR task, LibriSpeech-960h [48] corpus and GigaSpeech-M [49] with ∼480k training samples are utilized. For spoken QA task, we gather questions from a variety of sources, including Alpaca-52k [50], Web Questions [51], TriviaQA [52], SQuAD [53], Natural Questions [54], VoiceAssistant-400K from Mini-Omni [14] and UltraChat from SLAM-Omni [55]. Overall, there are ∼730k QA samples in our training datasets. For multi-round conversation, the conversation is generated by Llama-3-8B-Instruct based on a theme selected from TriviaQA and Natural Questions. Then the conversation is synthesized by CosyVoice2-0.5B. There are ∼80k multi-round conversation samples in our training datasets.

Moreover, we consider scenarios involving barge-ins and backchanneling in multi-turn conversations. The dataset includes two types of barge-ins. For context-independent barge-ins, we prompt GPT-4o to generate 10 direct interrupting sentences (*e.g. Please stop talking right now.*). For context-dependent barge-ins, we design a task to evaluate the model's ability to determine whether to stop speaking based on contextual cues. Specifically, we prompt Llama-3-8B-Instruct to generate questions that are either closely related to the ongoing dialogue (as meaningful interruptions) or entirely unrelated (as distractors). For backchanneling, we include 7 commonly used words or phrases (*e.g. Uh-huh.*). Notably, our experiments also take into account multi-speaker barge-in scenarios. More details about our training dataset can be found in Appendix A.

## 4.3 Evaluation

**Evaluation datasets and metrics.** We evaluate the speech interaction ability on four commonly used datasets, Llama Questions [56], Web Questions [51], TriviaQA [52] and AlpacaEval from Voicebench [57]. For Web Questions, the questions are converted into speech using a commercial TTS model from Volcano Engine. For triviaQA, we use the subset with 1000 samples from OpenAudioBench [27]. For knowledge QA datasets, Llama Questions, Web Questions and TriviaQA, accuracy are calculated as evaluation metric. For oral conversation dataset AlpacaEval, GPTScore by GPT-4.1-2025-04-14 is reported to score the appropriateness and reasonableness of the answer, ranging from 1 to 5 with 1 indicating the worst and 5 the best. Both answer text and speech are evaluated, noted as S2T and S2S. For S2S, answer speech is transcribed into text using Whisper-large-v3 first. The quality of the response speech is evaluated using the widely used UTMOS [58].

For evaluating full-duplex modeling capabilities, first, we report the success rate of turn-taking timing prediction, which reflects the model's ability to determine the appropriate moment to begin speaking. Second, we assess the model's ability to decide whether it should be interrupted, measured by the F1 score. For the context-independent setting, we collect 100 direct interrupting sentences – 50 spoken by the user and 50 by a third speaker – as positive examples, and 100 user-spoken backchanneling phrases as distractors. In the context-dependent setting, we collect 100 contextually relevant questions intended to interrupt the model (again, 50 from the user and 50 from a third speaker), while using 50 unrelated questions spoken by a third speaker and 50 silent samples as negative examples. Noted that the sentences used for evaluation are different from those during training.

To assess turn-taking statistics and real-world latency, we conduct live human-machine conversations across three scenarios: multi-turn dialogue, contextual-independent barge-in and contextual-dependent barge-in. A total of 60 dialogue sessions are collected (20 for each scenario). On average, conversations in multi-turn scenario consist of 5 turns with a mean duration of 75 seconds, whereas the dialogues in barge-in settings are shorter, averaging 3 turns. Statistics of turn-taking events including Inter-Pausal Unit (IPU), Pause and Gap are only evaluated in the multi-turn scenario.

**Baselines.** We select a large scale of performant open-source half-duplex or full-duplex speech LLMs as baselines, including Moshi [11], Freeze-Omni [20], GLM-4-Voice [15], Qwen2.5-Omni [22], VITA 1.5 [19], miniCPM-o [26], Kimi-Audio [21] and Baichuan-Audio [27]. All models are tested in an "oracle turn-taking" setting, in which models are forced to start speaking at the end of the question, except Moshi since it's difficult to force Moshi speak. Moreover, we also experiment with a more practical "predicted turn-taking" setting, in which the model needs to determine when to

start speaking by itself, which only full-duplex models can achieve. Moshi and Freeze-Omni serve as baselines in this setting.

# 5 Experimental Results

## 5.1 Comparison between implicit "thinking" and explicit "thinking"

We first compare the models trained with implicit and explicit "thinking" strategies in stage 1. As shown in Table 2, explicit "thinking" consistently outperforms implicit "thinking". We attribute this to training the LLM to output sequences containing both normal responses and state-transition-related tokens better aligns with its autoregressive nature. In contrast, the implicit "thinking" lacks complete feedback from the output back to the input of the LLM. An interesting observation is that after Stage 1, models achieve only around 70% turn-taking success rates on AlpacaEval. However, with the introduction of the assistant stream in Stage 2, success rates increase to approximately 90%, which highlights the importance of incorporating the assistant stream in full-duplex modeling for more robust turn-taking prediction. According to above results, we adopt explicit "thinking" as full-duplex strategy throughout this work. Actually, some variants can be derived from these two basic strategies, which are further analyzed in the Appendix C.

Table 2: Comparison of implicit and explicit thinking strategy.

| Full-duplex strategy | Llama Q. | | Web Q. | | TriviaQA | | AlpacaEval | |
|---|---|---|---|---|---|---|---|---|
| | succ. % | S2T | succ. % | S2T | succ. % | S2T | succ. % | S2T |
| Implicit | 97.3 | 76.3 | **99.9** | 50.8 | **95.8** | 62.0 | 67.3 | 3.73 |
| Explicit | **100.0** | **82.0** | **99.9** | **52.4** | 92.3 | **67.5** | **68.3** | **4.48** |

## 5.2 Comparison of the speech interaction capabilities between different speech LLMs

The main results of speech interaction abilities evaluation are shown in Table 3. Under the predicted turn-taking setting, SALMONN-omni achieves SOTA performance across all datasets, delivering an average 35.9% relative improvement over previous open-source full-duplex speech interaction LLMs. This substantial gain highlights SALMONN-omni's strong capabilities in handling a wide range of conversational scenarios—from general knowledge queries to everyday casual dialogue. In the oracle turn-taking setting, we further compare SALMONN-omni with more recent turn-based speech interaction LLMs. As illustrated in Table 3, SALMONN-omni achieves the best overall performance across all models, with best speech-to-text performances across 4 datasets and best speech-to-speech performances on Web Questions and TriviaQA. The average UTMOS across all datasets indicates that the speech generated by SALMONN-omni is of high quality. These results demonstrate that SALMONN-omni not only excels in speech understanding but also maintains highly stable and coherent speech generation.

## 5.3 Evaluation of full-duplex modeling capabilities

### 5.3.1 Turn-taking prediction

The turn-taking performances of different full-duplex speech interaction LLMs are presented in Table 4. The results show that SALMONN-omni achieves the highest success rate in accurately predicting turn-taking moments, averaged across multiple datasets. Notably, SALMONN-omni maintains strong performance even when handling short and fast-paced speech inputs, such as those in TriviaQA and AlpacaEval, a scenario in which Moshi and Freeze-Omni suffer a significant performance decline.

### 5.3.2 Distinguish true barge-ins from false barge-ins and backchanneling

**Comparison between different models on contextual-independent setting.** As shown in Table 5, SALMONN-omni demonstrates more robust performance on handling contextual-independent barge-ins and backchanneling. Moreover, both SALMONN-omni and Moshi can perform well when listening to their own echoes simultaneously, while Freeze-Omni exhibits a significant performance drop. It highlights the drawback of not implementing full-duplex models in a standalone manner.

Table 3: The performance of different speech interaction LLMs. For Llama Questions, Web Questions and TriviaQA, the evaluation metric is Acc.% For AlpacaEval, the evaluation metric is GPTScore. S2T is the speech-to-text performance and S2S is the speech-to-speech performance. The average UTMOS across all datasets is also reported with a more detailed version in Table 13.

| Model | Llama Q. | | Web Q. | | TriviaQA | | AlpacaEval | | Avg. |
|---|---|---|---|---|---|---|---|---|---|
| | S2T | S2S | S2T | S2S | S2T | S2S | S2T | S2S | UTMOS |
| | *Predicted turn-taking* | | | | | | | | |
| Moshi [11] | 60.8 | 54.5 | 23.4 | 22.1 | 25.6 | 16.7 | 1.84 | 1.76 | 3.178 |
| Freeze-Omni [20] | 74.2 | 56.2 | 40.8 | 27.9 | 45.1 | 28.5 | 3.90 | 2.46 | **4.277** |
| SALMONN-omni | **79.3** | **73.6** | **49.7** | **43.7** | **63.6** | **56.0** | **4.01** | **3.22** | 4.255 |
| | *Oracle turn-taking* | | | | | | | | |
| GLM-4-Voice [15] | 75.0 | 65.7 | 38.5 | 37.0 | 50.8 | 47.5 | 3.82 | **3.58** | 4.081 |
| Qwen2.5-Omni [22] | 78.7 | **75.7** | 41.9 | 39.6 | 55.5 | 52.2 | 3.04 | 2.61 | 4.085 |
| VITA-1.5 [19] | 77.3 | 56.0 | 43.9 | 31.0 | 54.8 | 41.3 | 3.92 | 1.94 | 4.022 |
| miniCPM-o [26] | 76.3 | 70.0 | 47.1 | 42.7 | 65.4 | 56.0 | 3.99 | 3.44 | 4.147 |
| Kimi-Audio [21] | 79.7 | 68.3 | 44.0 | 37.3 | 63.6 | 51.2 | 3.92 | 2.95 | 2.679 |
| Baichuan-Audio [27] | 76.0 | 74.0 | 43.1 | 40.7 | 56.9 | 53.0 | 3.76 | 3.32 | 4.182 |
| Freeze-Omni [20] | 77.0 | 61.0 | 42.3 | 29.1 | 53.5 | 36.5 | 3.71 | 2.42 | **4.276** |
| SALMONN-omni | **80.0** | 75.0 | **50.5** | **45.4** | **66.0** | **58.8** | **4.05** | 3.33 | 4.261 |

Table 4: Turn-taking success rate (%) of full-duplex speech interaction LLMs.

| Model | Llama Q. | Web Q. | TriviaQA | AlpacaEval |
|---|---|---|---|---|
| Moshi | 85.0 | 76.0 | 37.1 | 83.4 |
| Freeze-Omni | **99.7** | **99.8** | 72.0 | 87.9 |
| SALMONN-omni | **99.7** | 99.6 | **92.8** | **92.0** |

Since Freeze-Omni relies on an additional VAD module, its performance is severely affected when the VAD module fails to provide accurate speech segmentation.

Table 5: Comparison between different models on contextual-independent barge-ins and backchanneling. Echo factor means the model can hear $\times n$ its own echo.

| Model | Echo factor | Precision | Recall | F1 score |
|---|---|---|---|---|
| Moshi | $\times 1.0$ | **0.84** | 0.77 | 0.80 |
| Freeze-Omni | $\times 0.0$ | 0.64 | 0.72 | 0.68 |
| Freeze-Omni | $\times 0.1$ | 0.19 | 0.86 | 0.31 |
| Freeze-Omni | $\times 1.0$ | 0.10 | 0.69 | 0.17 |
| SALMONN-omni | $\times 1.0$ | **0.84** | **0.92** | **0.88** |

**DPO further enhances model's full-duplex modeling capabilities.** We train SALMONN-omni on both context-independent and context-dependent settings, and it turns out that the model can handle both scenarios effectively, which demonstrates our framework can handle complex context-aware dynamics in free-form conversations. However, after the two SFT stages, the low precision shows that SALMONN-omni equips a tendency to be interrupted, which reflects the model's insufficient understanding of context. Thus, we incorporate DPO for post-training the model, and the overall performance improvement confirms the effectiveness of this approach. As shown in Fig. 4 and Table 6, an interesting phenomenon observed during DPO training is that the model initially becomes extremely conservative, rarely getting interrupted at all. However, as training progresses, the model's capabilities gradually recover and eventually surpass those of the SFT-trained models. More details can be found in Appendix D.

Table 6: The performance of SALMONN-omni on contextual-dependent and contextual-independent settings through the DPO training stage with batch size set to 256.

| Stage | Contextual-Independent | | | Contextual-Dependent | | | Overall |
|---|---|---|---|---|---|---|---|
| | Precision | Recall | F1 score | Precision | Recall | F1 score | F1 score |
| SFT (Stage 2) | 0.68 | 0.93 | 0.79 | 0.88 | 0.99 | 0.93 | 0.86 |
| DPO - 10 steps | 0.88 | 0.07 | 0.13 | 0.95 | 0.21 | 0.34 | 0.24 |
| DPO - 40 steps | 0.84 | 0.92 | **0.88** | 0.89 | 0.98 | 0.93 | **0.90** |

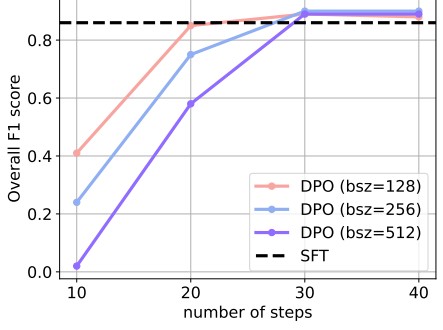

Figure 4: The overall F1 score of SALMONN-omni when trained with different batch sizes during the DPO stage.

Table 7: Statistics of turn-takings events (IPU, Pause and Gap) in real-world human-machine dialogues.

| | IPU | Pause | Gap |
|---|---|---|---|
| Ground Truth | 61.89s | 5.49s | 6.50s |
| SALMONN-omni | 63.30s | 4.68s | 5.01s |

### 5.4 Turn-taking statistics and latency evaluation in real-world deployment

The turn-taking statistics in the multi-turn dialogue scenario (normalized to 5-turn dialogues) are shown in Table 7, which is largely consistent with the statistical properties of the training data. As for latency, SALMONN-omni exhibits an average Time-to-First-Audio (TTFA) of 405ms. This latency is primarily a consequence of its design, which mandates waiting for four time blocks to accumulate sufficient context for the speech synthesizer. We posit that this latency is not a fundamental limitation and could be substantially reduced through synthesizer optimization. In contrast, the model demonstrates its capacity for rapid interaction in the barge-in scenarios, achieving response latencies of approximately 128 ms and 144 ms for contextual-independent and contextual-dependent interruptions, respectively. These results confirm the model's ability to react swiftly to user interjections.

## 6 Conclusion

In this work, we present SALMONN-omni, the first standalone full-duplex speech LLM that operates without injecting audio codec tokens into its vocabulary. By seamlessly integrating a streaming speech encoder, a unified LLM backbone, and a streaming speech synthesiser, and introducing a novel explicit "thinking" strategy for predicting dialogue state transitions, SALMONN-omni autonomously learns when to listen and when to speak. Extensive experiments on standard benchmarks for spoken QA and open-domain dialogue show that SALMONN-omni achieves an average 35.9% relative accuracy improvement over existing open-source full-duplex models in predicted turn-taking scenarios, while remaining highly competitive with half-duplex models trained on significantly larger datasets. Furthermore, SALMONN-omni demonstrates robust modeling of full-duplex conversational dynamics, including turn-taking, backchanneling, echo cancellation, and context-dependent barge-ins. Additional RL further enhances the model's ability to handle real-time dialogue behaviours with improved fluency and responsiveness.

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

# A Further experimental details

## A.1 Details for training data

For VoiceAssistant-400K and UltraChat, we only select the first-round QA and filter repeated questions. We utilize the question speech in the dataset and re-synthesize answer speech based on the text answer by CosyVoice2-0.5B [46] for training. For other dataset, text responses are generated by Llama-3-8B-Instruct and both question and speech audio are synthesized by CosyVoice2-0.5B.

Table 8: Training dataset details.

| Task | Dataset | #Samples |
|---|---|---|
| ASR | LibriSpeech | 281k |
| | GigaSpeech | 200k |
| QA | Alpaca-52k | 39k |
| | Web Questions | 4k |
| | TriviaQA | 58k |
| | SQuAD | 127k |
| | Natural Questions | 301k |
| | VoiceAssistant-400k | 79k |
| | UltraChat | 120k |
| multi-round conversation | TriviaQA | 25k |
| | Natural Questions | 56k |

## A.2 Training specifications

The Mamba streaming encoder is pre-trained on LibriHeavy [59] and GigaSpeech [49] for 300k steps with a batch size of 512. The AdamW [60] optimizer with $\beta_1 = 0.9$ and $\beta_2 = 0.95$ were used. In the first two stages, the batch size is set to 128 and the learning rates are $4 \times 10^5$ and $3 \times 10^5$ respectively. In the third stage, batch sizes include 128, 256 and 512 are compared with learning rate set to $1 \times 10^6$. All training processes are performed on 32 A100 GPUs with 50k and 30k steps for stage 1 and 2. Model checkpoints with the best validation accuracy are used for evaluation.

# B Prompts

> **Prompts for Llama-3-8B-Instruct to generated spoken QA response**
>
> Please answer the following question {} in a conversational style. Keep your response concise, to the point, and within 300 tokens.

> **Prompts for Llama-3-8B-Instruct to generated multi-turn dialogues**
>
> Based on the topic: {}. Write the transcript of a conversation between two people A and B. Use short turns. The conversation should be in spoken form (especially the numbers or dates should be in spoken form) and doesn't include ∗xxx∗ or [xxx] or (xxx).

**Prompts for Llama-3-8B-Instruct to generated context-dependent barge-in data**

The following is a conversation between A and B.

{}

Please ask B a question based on the conversation. This question needs to be strongly related to what B said. The question should be concise, short and in spoken form. Please only output your response.

**Prompts for GPT-4.1-2025-04-14 to evaluate oral conversation results on AlpacaEval**

I need your help to evaluate the performance of several models in the speech interaction scenario. The models will receive a speech input from the user, which they need to understand and respond to with a speech output.

Your task is to rate the model's responses solely based on their accuracy and reasonableness, using the provided user input transcription [Instruction] and the model's output transcription [Response]. Please evaluate the response on a scale of 1 to 5:

1 point: The response is clearly incorrect, illogical, or entirely fails to address the user's request.

2 points: The response contains major inaccuracies or flaws in reasoning, even if it appears somewhat related to the question.

3 points: The response is mostly reasonable and factually correct but may include minor mistakes or unsupported assumptions.

4 points: The response is accurate and logically sound, with only negligible or borderline issues.

5 points: The response is fully accurate and logically consistent. It reflects a correct and well-reasoned understanding of the user's input.

Below are the transcription of user's instruction and models' response:
### [Instruction]
{}

### [Response]
{}

After evaluating, please output the score only without anything else. You don't need to provide any explanations.

**Prompts for gpt-4o-audio-preview-2024-12-17 to evaluate emotion intensity**

Given a speech audio clip, evaluate whether the speaker expresses any emotion, and if so, how strongly and clearly that emotion is conveyed. Focus on ∗vocal features∗ such as pitch variation, intonation, rhythm, volume, and prosodic dynamics. Do ∗not∗ infer from semantic content — only judge the presence and intensity of emotion in how the ∗voice sounds∗. Output a single integer score from 1 to 5 based on the following scale: ### Scoring Rubric:

∗∗1 — No Emotion∗∗: Completely flat and monotone. Voice sounds like a neutral, mechanical reading or a dry recitation.

∗∗2 — Barely Emotional∗∗: Emotion may feel suppressed or accidental. Difficult to confidently say any emotion is present.

∗∗3 — Mild Emotion∗∗: Emotion is noticeably present but limited in strength. Vocal delivery includes some clear variation in tone or rhythm suggesting emotional engagement.

∗∗4 — Clear Emotion∗∗: Emotion is clearly conveyed through dynamic changes in pitch, volume, and timing. Voice reflects emotional involvement (e.g., excitement, frustration, warmth).

∗∗5 — Strong and Expressive Emotion∗∗: Emotion is vivid, intense, and unmistakable. Rich, nuanced control over prosody (e.g., expressive pitch arcs, pauses, stress) highlights emotional intensity.

After evaluating, please output the score only without anything else. You don't need to provide any explanations.

# C  Other Variants of Thinking Strategy

We also explore other thinking strategies with different thinking contents. We experiment with a small dataset of only LibriSpeech and Alpaca-52k. The ASR results are reported in Table 9.

**Implicit-ASR.**  Implicit-ASR is a variant of the implicit "thinking" strategy. Here we only use one special token `<shift>`. At those steps to keep silent, the model is trained to perform streaming ASR. Moreover, within those time blocks that the model has finished the text response and is waiting for speech synthesizing, we utilize an LLM to generate some inner thoughts related to the current conversation as the label. At those points for state transition, it should output `<shift>` marking the state transition.

**Explicit-ASR.**  Explicit-ASR is similar to Implicit-ASR, apart from that the thinking contents are sent back to the input autoregressively.

**Explicit-NS.**  Explicit-NS has two special tokens `<think>` and `<shift>`. For the time blocks without generating text response, `<think>` is used as the input and `<shift>` is used as the label. However, the weight of the loss is negative except for those time blocks in which the model needs to perform state transition.

Table 9: The performance of different "thinking" strategies on ASR task. succ. is the success rate of turn-taking prediction.

| "Thinking" Strategy | test-clean | | test-other | |
| --- | --- | --- | --- | --- |
| | succ. | %WER ($\downarrow$) | succ. | %WER ($\downarrow$) |
| Implicit | 93.3 | 2.67 | 94.6 | 6.63 |
| Implicit-ASR | 94.7 | 3.48 | **95.7** | 8.26 |
| Explicit | **95.6** | **2.40** | 95.6 | **6.09** |
| Explicit-ASR | 91.8 | 5.09 | 94.2 | 9.85 |
| Explicit-NS | 91.8 | 7.65 | 92.9 | 10.5 |

As shown in 9, increasing the diversity of labels during the thinking process (listening state) hurts performance. We assume that the performance differences of different thinking labels may be partially attributed to the full-duplex finetuning. In a text LLM, the model doesn't need to generate explicit responses upon receiving user input. However, for full-duplex speech interaction LLMs, the model must consistently generate responses to manage turn-taking in natural dialogues, alternating between speaking and listening states. During full-duplex fine-tuning, the model must learn two critical skills: first, the ability to model state transitions; second, the ability to distinguish between the operational modes of the speaking and listening states. When using the simple Explicit thinking strategy, the listening state (where the model generates only a single special token) is easily distinguishable. This allows the model to focus on learning state transitions and optimizing the speaking state's operation mode, leading to high turn-taking precision and improved content quality. However, as the diversity of thinking labels increases, the operation modes of the two states become less distinct, which sometimes results in undesirable messy generation during the speaking state. Meanwhile, while oracle "thinking" ASR content can enhance speech generation, incorrect "thinking" content during inference may introduce biases and hurt the quality of generated speech due to the error propagation in response generation.

# D  More Results on DPO Post-Training for Full-duplex Modeling

As shown in Table 10, 11, 12, the intriguing observation, consistent across different training batch sizes, is that during DPO training, the model initially becomes overly cautious—rarely getting interrupted. However, as training progresses, its performance steadily improves and eventually surpasses that of the SFT-trained model.

Table 10: The performance of SALMONN-omni on contextual-dependent and contextual-independent settings through the DPO training stage with batch size set to 128. (detailed version)

| Stage | Contextual-Independent | | | Contextual-Dependent | | | Overall |
|---|---|---|---|---|---|---|---|
| | Precision | Recall | F1 score | Precision | Recall | F1 score | F1 score |
| SFT (Stage 2) | 0.68 | 0.93 | 0.79 | 0.88 | 0.99 | 0.93 | 0.86 |
| DPO - 10 steps | 0.88 | 0.14 | 0.24 | 0.97 | 0.38 | 0.55 | 0.41 |
| DPO - 20 steps | 0.78 | 0.75 | 0.77 | 0.91 | 0.96 | **0.94** | 0.85 |
| DPO - 30 steps | 0.83 | 0.88 | **0.85** | 0.90 | 0.98 | 0.93 | **0.89** |
| DPO - 40 steps | 0.81 | 0.84 | 0.82 | 0.90 | 0.95 | 0.93 | 0.88 |

Table 11: The performance of SALMONN-omni on contextual-dependent and contextual-independent settings through the DPO training stage with batch size set to 256. (detailed version)

| Stage | Contextual-Independent | | | Contextual-Dependent | | | Overall |
|---|---|---|---|---|---|---|---|
| | Precision | Recall | F1 score | Precision | Recall | F1 score | F1 score |
| SFT (Stage 2) | 0.68 | 0.93 | 0.79 | 0.88 | 0.99 | 0.93 | 0.86 |
| DPO - 10 steps | 0.88 | 0.07 | 0.13 | 0.95 | 0.21 | 0.34 | 0.24 |
| DPO - 20 steps | 0.92 | 0.45 | 0.60 | 0.98 | 0.78 | 0.87 | 0.75 |
| DPO - 30 steps | 0.84 | 0.88 | 0.86 | 0.92 | 0.97 | **0.95** | **0.90** |
| DPO - 40 steps | 0.84 | 0.92 | **0.88** | 0.89 | 0.98 | 0.93 | **0.90** |

## E  Speech Quality Assessment

Detailed UTMOS on all the datasets of each model is reported in Table 13.

## F  Emotional Speech Generation

We also explore equipping SALMONN-omni with emotional speech generation to support more natural and human-like speech interactions. To achieve this, the LLM backbone is prompted to produce an emotion indicator prior to generating its verbal response, and the speech synthesizer operates in an instruction TTS style. Emotional speech samples are synthesized using a commercial TTS model provided by Volcano Engine. The emotion intensity of the emotion enhanced SALMONN-omni, along with baseline models, is evaluated on an in-house dataset using gpt-4o-audio-preview-2024-12-17, with results summarized in Table 14. Findings show that SALMONN-omni achieves the highest emotion intensity among all evaluated systems. However, its emotional expression remains inconsistent, occasionally producing responses with inappropriate or mismatched emotions.

## G  Limitation

While SALMONN-omni is a high-performance full-duplex speech interaction LLM, it still faces several limitations. First, although it effectively predicts turn-taking and can distinguish between user barge-ins and backchanneling, it rarely generates backchanneling responses during user speech, primarily due to limitations in the training data. Enhancing this capability is essential for achieving more natural, human-like conversational behavior. Second, SALMONN-omni's training sequence is limited to three minutes due to memory constraints. As a result, its performance noticeably degrades when interaction exceeds this duration. Future work involves enabling long-term memory for full-duplex speech interaction LLM.

## H  Impact Statement

This paper aims to explore robust full-duplex speech interaction framework, advancing the frontier of real-time, fully interactive conversational AI. We believe that, alongside enhancing the performance of conversational AI, it is essential to prioritize AI safety. This includes ensuring that all generated content is free from harm, discrimination, and bias, as well as developing detection models to identify

Table 12: The performance of SALMONN-omni on contextual-dependent and contextual-independent settings through the DPO training stage with batch size set to 512. (detailed version)

| Stage | Contextual-Independent | | | Contextual-Dependent | | | Overall |
| --- | --- | --- | --- | --- | --- | --- | --- |
| | Precision | Recall | F1 score | Precision | Recall | F1 score | F1 score |
| SFT (Stage 2) | 0.68 | 0.93 | 0.79 | 0.88 | 0.99 | 0.93 | 0.86 |
| DPO - 10 steps | 1.0 | 0.01 | 0.02 | 1.0 | 0.01 | 0.02 | 0.02 |
| DPO - 20 steps | 0.88 | 0.29 | 0.44 | 0.98 | 0.96 | 0.55 | 0.58 |
| DPO - 30 steps | 0.82 | 0.86 | 0.84 | 0.93 | 0.97 | **0.95** | **0.89** |
| DPO - 40 steps | 0.82 | 0.88 | **0.85** | 0.91 | 0.95 | 0.93 | **0.89** |

Table 13: UTMOS of SALMONN-omni and other speech LLMs on spoken QA task.

| Model | Llama Q. | Web Q. | TriviaQA | AlpacaEval |
| --- | --- | --- | --- | --- |
| | *Predicted turn-taking* | | | |
| Moshi [11] | 3.245 | 3.257 | 3.017 | 3.193 |
| Freeze-Omni [20] | 4.288 | 4.273 | 4.293 | 4.255 |
| SALMONN-omni | 4.256 | 4.250 | 4.272 | 4.243 |
| | *Oracle turn-taking* | | | |
| GLM-4-Voice [15] | 4.180 | 4.107 | 4.145 | 3.890 |
| Qwen2.5-Omni [22] | 4.140 | 4.124 | 4.127 | 3.950 |
| VITA-1.5 [19] | 4.042 | 4.091 | 4.104 | 3.852 |
| miniCPM-o [26] | 4.171 | 4.152 | 4.155 | 4.110 |
| Kimi-Audio [21] | 2.882 | 2.713 | 2.793 | 2.327 |
| Baichuan-Audio [27] | 4.261 | 4.216 | 4.292 | 3.959 |
| Freeze-Omni [20] | 4.300 | 4.281 | 4.278 | 4.245 |
| SALMONN-omni | 4.262 | 4.263 | 4.279 | 4.240 |

AI-generated content. It is also crucial that all users are made aware when they are interacting with a conversational AI chatbot.

# I Case Studies

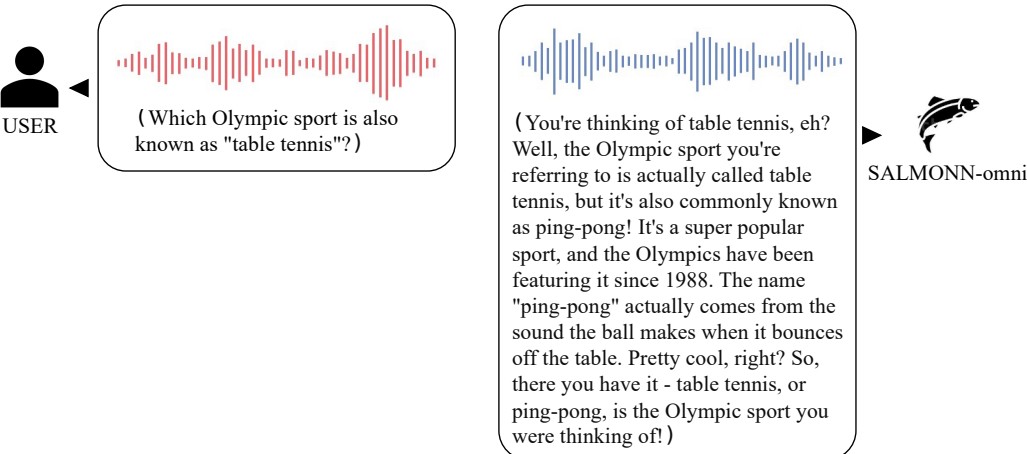

Figure 5: Spoken QA: SALMONN-omni can handle turn-taking in spoken question answering scenarios with "thinking" mechanism.

Table 14: Emotion intensity of speech interaction LLMs.

| Model | Emotion intensity |
|---|---|
| GLM-4-Voice | 3.30 |
| Qwen2.5-Omni | 3.27 |
| VITA-1.5 | 3.03 |
| miniCPM-o | 3.16 |
| Kimi-Audio | 3.39 |
| Baichuan-Audio | 3.14 |
| Freeze-Omni | 3.00 |
| SALMONN-omni | **3.49** |

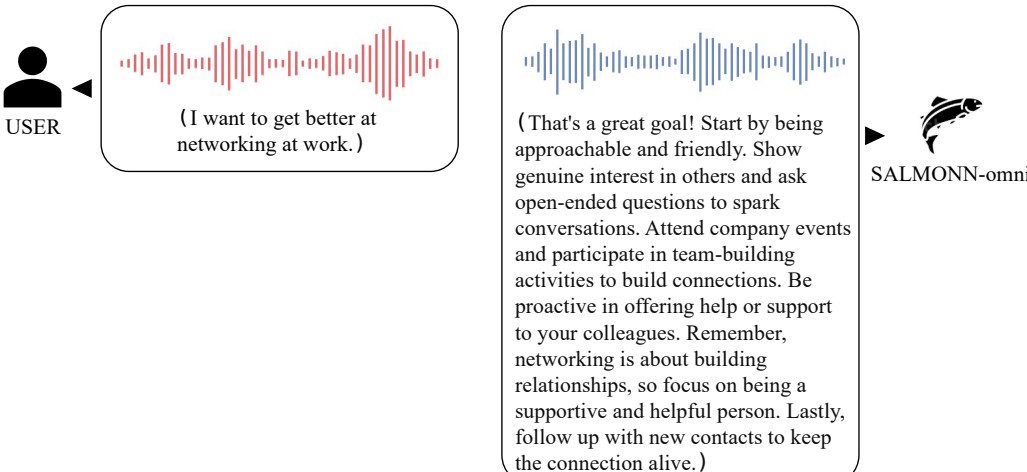

Figure 6: Open-domain dialogue: SALMONN-omni can handle turn-taking in spoken question answering scenarios with "thinking" mechanism.

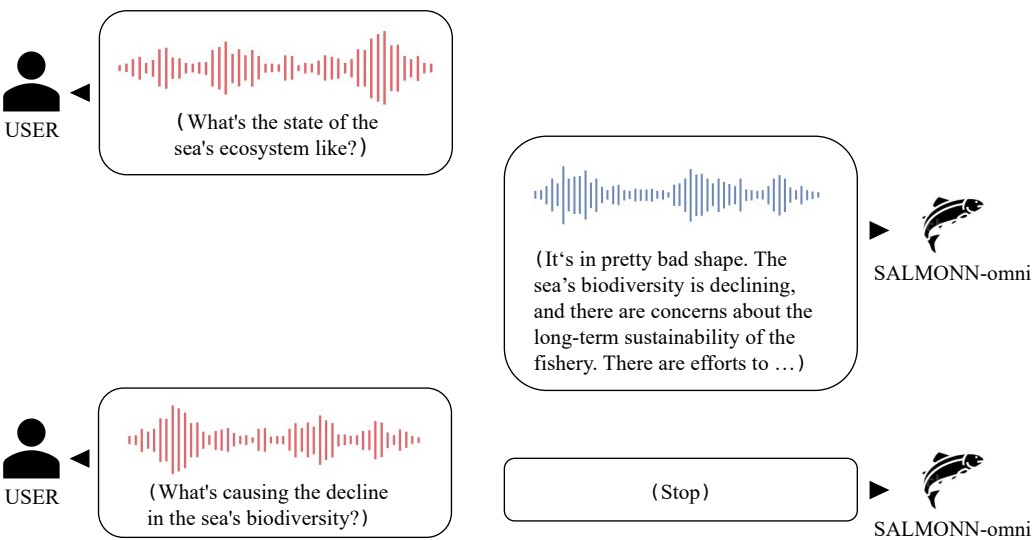

Figure 7: Context-dependent barge-in: When the user is quick to buzz in with the response, SALMONN-omni can stop generating speech.

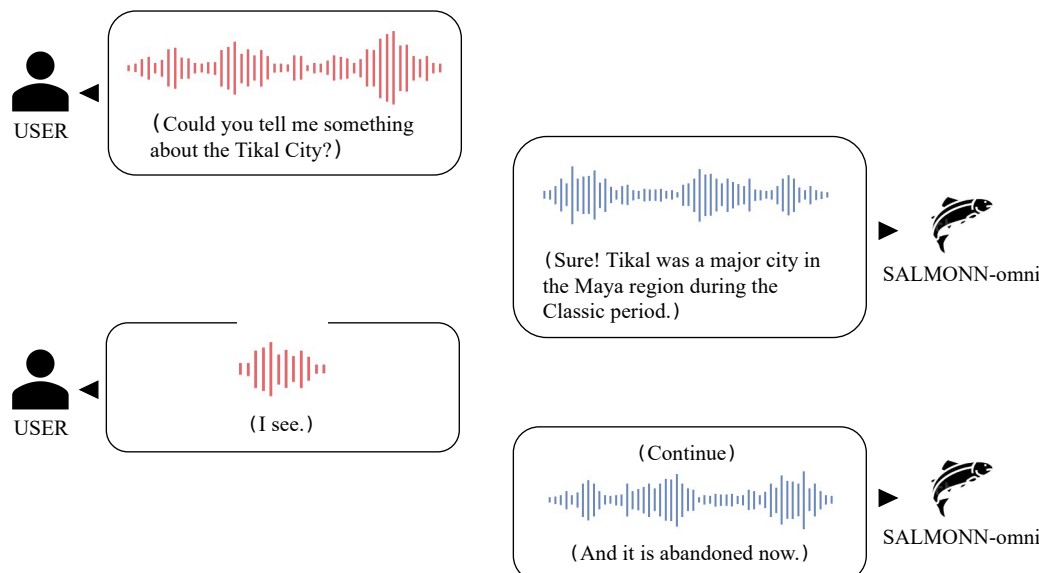

Figure 8: Distinguishing backchanneling: When the user interjects with a backchannel cue, SALMONN-omni can continue speaking.

