# OpenReview forum: "SALMONN-omni: A Standalone Speech LLM without Codec Injection for Full-duplex Conversation"
_NeurIPS.cc/2025/Conference — NeurIPS 2025 poster_

### Official Review · Reviewer_tsTQ · 2025-06-29

**Clarity:** 2
**Significance:** 3
**Originality:** 3
**Rating:** 5
**Confidence:** 4

**Summary:**

This paper proposes SALMONN-omni, a full-duplex speech LLM that achieves speech-text modality fusion and natural conversational interaction without compromising LM capabilities. The framework integrates a streaming speech encoder and synthesizer into the LLM backbone, enabling simultaneous processing of environmental audio and assistant speech. An explicit "thinking" strategy and subsequent DPO optimization enhance duplex interaction abilities. Experiments on spoken QA and open-domain dialogue validate the model’s duplex capabilities.

**Questions:**

- Q1 Compared to the backbone LLM, does the final model exhibit degradation in reasoning/understanding due to added modules (LoRA, encoder adapter, etc.)
- Q2 For the streaming speech encoder: Why choose Mamba blocks?
- Q3 Lines 205–206 (Stage 3): Could you clarify the conclusion where the model "exhibits a clear bias toward being interrupted and shows insufficient understanding of dialogue semantics"?
- Q4 For DPO, it's better to detail the data scale, construction process, and reward design criteria for better understanding.
- Q5 Line 232–233: For synthesized conversation data, how were speaker attributes (timbre/accent/emotion) selected in the generation process?
- Q6 This question does not affect scoring. Existing works in the field of speech generation, especially in speech dialogue, often employ different pre-training models and dataset with different designed complexity. How to compare more fairly in the research, and how to better identify the effectiveness of the proposed method.

**Ethical Concerns:**

["NO or VERY MINOR ethics concerns only"]

**Final Justification:**

The author addressed my concerns one by one in the discussion, and 5 (Accept) can be given after the author's stated revisions to the paper.

**Limitations:**

yes

**Quality:**

3

**Strengths And Weaknesses:**

Strengths
- S1: A well-executed study: Efficiently enables end-to-end speech via encoder/synthesizer integration while freezing the LLM backbone. The explicit state strategy reduces duplex system complexity, and DPO further boosts performance.
- S2: Comprehensive experiments covering duplex dynamics, QA, daily dialogue, and ablation studies on state transition strategies.

Weaknesses
- W1: Some methodological details and experimental setups could be clearer.
- W2: Lacks evaluation of generation quality (e.g., subjective/objective metrics for speech quality).

---

> ### Author Rebuttal · Authors · 2025-07-31
>
> We sincerely appreciate the high score you have given to our work. We hope the following content will further address and alleviate any remaining concerns you may have.
>
> - **Q1: Some methodological details and experimental setups could be clearer.**
>
> Thank you for your suggestions. We will improve the presentation of the paper accordingly. Several experimental setups are detailed in the Appendix: Training data and specifications are in Appendix A; Prompts used for generating the training data and ChatGPT Score evaluation are in Appendix B. If the paper is accepted, we also plan to release our source code, training data, and pipeline to further clarify the implementation details. Please let us know if any additional aspects of our setup require clarification.
>
> - **Q2: Lacks evaluation of generation quality (e.g., subjective/objective metrics for speech quality).**
>
> Thanks for your valuable advice. Speech quality is also an important evaluation aspect for speech generation. Here we compute the UTMOS score (a widely used automatic MOS prediction model, also used by GLM-4-Voice, the score is between 1 and 5, the higher the better) for answer samples.
>
> |                  | Llama | Web   | Trivia | Alpaca |
> |------------------|-------|-------|--------|--------|
> | **Predicted turn-taking** |       |       |        |        |
> | moshi            | 3.245 | 3.257 | 3.017  | 3.193  |
> | freeze-omni      | 4.288 | 4.273 | 4.293  | 4.255  |
> | SALMONN-omni     | 4.256 | 4.250 | 4.272  | 4.243  |
> | **Oracle turn-taking**    |       |       |        |        |
> | GLM-4-Voice      | 4.180 | 4.107 | 4.145  | 3.890  |
> | Qwen2.5-Omni     | 4.140 | 4.124 | 4.127  | 3.950  |
> | VITA-1.5         | 4.042 | 4.091 | 4.104  | 3.852  |
> | miniCPM-o        | 4.171 | 4.152 | 4.155  | 4.110  |
> | Kimi-Audio       | 2.882 | 2.713 | 2.793  | 2.327  |
> | Baichuan-Audio   | 4.261 | 4.216 | 4.292  | 3.959  |
> | Freeze-Omni      | 4.300 | 4.281 | 4.278  | 4.245  |
> | SALMONN-omni     | 4.262 | 4.263 | 4.279  | 4.240  |
>
> The results show that SALMONN-omni can generate high-quality speech, with average UTMOS quality around 4.25 (For MOS score, 4 means good, 5 means excellent). It indicates SALMONN-omni can not only generate speech faithfully (as shown in high S2S performance), but also generate high-quality speech.
>
> - **Q3: Compared to the backbone LLM, does the final model exhibit degradation in reasoning/understanding due to added modules (LoRA, encoder adapter, etc.)**
>
> We compared the performance of our model using both text and audio inputs with the original performance of LLaMA-3. It turns out that the performance degradation of the LLM is not significant, and our model exhibits comparable performance when using text ("Ours-text") and audio inputs ("Ours-wav"), which suggests that avoiding codec injection into the LLM indeed results in only minimal degradation to the model’s performance. Notably, we did not incorporate any pure text data during training. Such an approach is beneficial for preserving LLM performance in works like Qwen2.5-Omni. [1]
>
> ||Llama Q.|Web Q.|Trivia Q.|AlpacaEval|
> |---|---|---|---|---|
> |Llama-3|0.85|0.56|0.73|3.83|
> |Ours-text|0.84|0.52|0.68|3.82|
> |Ours-wav|0.80|0.51|0.66|4.05|
>
> [1] Jin Xu, Zhifang Guo, Jinzheng He, et al. “Qwen2.5-Omni technical report”. In: arXiv preprint arXiv:2503.20215 (2025).
>
> - **Q4: For the streaming speech encoder: Why choose Mamba blocks?**
>
> It is an alternative design choice and  not the primary focus of this work. One can definitely select any architecture that can support streaming speech processing, such as CNN and casual transformer. We choose Mamba since Mamba can support streaming sequence modeling with a relatively low computational cost compared to the casual transformer and it shows reasonable performance in our preliminary experiments.
>
> - **Q5: Lines 205–206 (Stage 3): Could you clarify the conclusion where the model "exhibits a clear bias toward being interrupted and shows insufficient understanding of dialogue semantics"?**
>
> This is an empirical observation: we found that after Stage 2, the model exhibited high recall but relatively low accuracy on conversational dynamics evaluations (where we consider being interruptible as a positive case). We believe this is primarily because the model determines whether it is being interrupted, not based on semantic understanding, but rather on the presence of another speaker’s voice. The model tends to prefer being interrupted because the resulting output sequence is shorter, which may lead to a higher overall probability for the complete sentence.
>
> - **Q6: For DPO, it's better to detail the data scale, construction process, and reward design criteria for better understanding.**
>
> Thank you for your suggestion. We clarify the composition of the DPO data here and will include these details in the next version. First, we obtained the Stage 2 model on the dialogue training data from Stage 2, collecting 11,000 samples involving errors in handling dynamics such as barge-in and backchanneling. These 11k samples evenly cover five categories of errors: incorrect handling of contextual-dependent barge-in, user-triggered contextual-independent barge-in, third-party-triggered contextual-independent barge-in, backchanneling, and early stop in the absence of any interruptions. For positive samples (with reward 1), we constructed them by directly modifying the position of the shift token in the negative samples (with reward 0), ensuring the accuracy of the reward signal. Moreover, we randomly sampled another 11k examples from other tasks within the Stage 2 dataset to prevent the model from forgetting knowledge related to those tasks. Using these 22k samples, we applied DPO loss to tasks related to barge-in and backchanneling, while continuing to use SFT for the remaining tasks.
>
> - **Q7: Line 232–233: For synthesized conversation data, how were speaker attributes (timbre/accent/emotion) selected in the generation process?**
>
> Our speaker prompts are selected from the LibriHeavy dataset. Although this dataset does not provide explicit speaker attribute annotations, we believe it offers a diverse range of speaker characteristics, as LibriHeavy was specifically designed to support the development of robust ASR models.
>
> - **Q8: This question does not affect scoring. Existing works in the field of speech generation, especially in speech dialogue, often employ different pre-training models and dataset with different designed complexity. How to compare more fairly in the research, and how to better identify the effectiveness of the proposed method.**
>
> It's a good question. And it's the fact that almost all comparisons between different speech dialogue models are not strictly fair, since different models utilize different datasets and training specifications (such as batch size, training steps and so on). Under different setups, it's hard to say where the preference gain comes from, maybe the training dataset, maybe the architecture design, maybe the training strategies. It's hard or even impossible to disentangle the contribution of each part. In one paper, we can do ablation studies to search for a better design. But we can not do ablation studies between papers. Open-sourcing training data and model weights is certainly a critical prerequisite for fair comparisons. While many company-developed models currently rely on proprietary data, we also advocate for researchers in the academic community to make their training data publicly available whenever possible.

---

> > ### Comment · Reviewer_tsTQ · 2025-08-06
> > **Response from Reviewer**
> >
> > Thanks for your detailed response. Your answers have addressed my concerns, and I will raise my score.

---

> > > ### Author Response · Authors · 2025-08-06
> > > **further response**
> > >
> > > We are very pleased that our response has alleviated some of your concerns. Thank you for your suggestions; they are extremely important for improving the quality of our paper.

---

### Official Review · Reviewer_2yL6 · 2025-07-02

**Clarity:** 3
**Significance:** 4
**Originality:** 4
**Rating:** 5
**Confidence:** 4

**Summary:**

This paper introduces SALMONN-omni, a novel full-duplex speech-language model designed for real-time, naturalistic spoken dialogue. Unlike prior systems that rely on codec token injection (e.g., Moshi) or modular dual-LLM architectures (e.g., Freeze-Omni), SALMONN-omni is a standalone codec-free solution that integrates a streaming speech encoder, a single LLM, and a speech synthesizer in an end-to-end trainable framework. The model handles simultaneous speech input/output and autonomously manages dialogue transitions through a new “thinking” strategy involving explicit state-transition tokens (e.g., <think>, <shift>).

The authors implement a three-stage training process: (1) encoder-LLM alignment, (2) speech synthesis integration, and (3) reinforcement learning (via DPO) for improved dialogue dynamics. Experimental results on speech QA, dialogue, and turn-taking benchmarks show a significant performance gain; up to 35.9% relative improvement over previous open-source full-duplex models. SALMONN-omni also achieves competitive results against strong half-duplex models trained on much larger corpora. The model further demonstrates robustness in backchannel detection, echo cancellation, and barge-in handling.

**Questions:**

1. Have you evaluated model behavior with spontaneous, noisy user inputs beyond TTS-generated samples?

2. Can you provide latency measurements under real-time deployment (e.g., wall-clock delay per turn)?

3. How would the model scale to multilingual or code-switched dialogue?

4. Could you share insights on where the model fails, especially in ambiguous barge-in situations?

**Ethical Concerns:**

["NO or VERY MINOR ethics concerns only"]

**Final Justification:**

The work is technically solid and all my concerns have been appropriately addressed.

**Limitations:**

- Evaluation uses synthetic speech inputs, which may not capture conversational noise, hesitation, or accent variation.

- While turn-taking is well-evaluated, the paper lacks a detailed ablation or error analysis that would provide insight into specific failure cases.

**Quality:**

4

**Strengths And Weaknesses:**

**Strengths:**

- **Novel architecture:** SALMONN-omni is the first open-source standalone full-duplex speech LLM without audio codec injection.

- **Well-motivated design:** The “thinking” strategy and stream interleaving offer a clean integration of speech and dialogue dynamics.

- **Competitive with large-scale models:** Despite being trained on far less data than models like Kimi-Audio, SALMONN-omni delivers strong results across QA and dialogue tasks.

- **RL-based modeling of conversation flow:** Application of DPO to barge-in and turn-taking tasks is novel and well-justified.

- **Comprehensive evaluation:** Model is tested across oracle and predicted turn-taking, S2T and S2S, and real-time conversational phenomena.

**Weaknesses:**

**Lack of human evaluation:** The paper reports GPTScore and ASR-based metrics but no human judgments for naturalness, fluency, or appropriateness of the dialogue.

**Synthetic evaluation conditions:** Many input utterances are generated via TTS and text prompts, which might not reflect real-world speech variability.

**No analysis of failure cases:** For example, in barge-in detection or speech overlap, it’s unclear where the model fails and how errors propagate.

---

> ### Author Rebuttal · Authors · 2025-07-31
>
> We sincerely thank the reviewer for the recognition of our work. To ensure clarity, we have summarized the reviewer’s comments into the following key points. We hope this presentation addresses your concerns. If you feel that any aspect has not been adequately captured, we would greatly welcome further discussion.
>
> - **Q1: Lack of human evaluation: The paper reports GPTScore and ASR-based metrics but no human judgments for naturalness, fluency, or appropriateness of the dialogue.**
>
> Thank you for your invaluable suggestions. We agree that incorporating human evaluation is highly important. However, due to time constraints, it is challenging to organize a rigorous human assessment. As a result, we have temporarily relied on the use of ChatGPT Score to evaluate the model’s conversational capabilities in real-world scenarios. In the following real-world scenario evaluations, we provide a detailed explanation of our evaluation methodology.
>
> - **Q2: Synthetic evaluation conditions: Many input utterances are generated via TTS and text prompts, which might not reflect real-world speech variability.**
>
> We sincerely appreciate the reviewer’s suggestion. In response, we deployed our model and evaluated its performance in real-world human-machine interaction scenarios. The experimental results are presented below.
>
> - **Q3: Latency measurements under real-time deployment**
>
> In the description of the real-time deployment experiments that follow, we evaluate the model’s turn-taking statistics, response latency to users, and latency in reacting to user interruptions.
>
> - **Q4: Insights on failure cases**
>
> Here we would like to share some humble insights on some failure situations of turn-taking and barge-ins.
>
> 1) In turn-taking scenarios, when the speech question sample is very quick and short, the model tends to fail to determine when to start speaking. As shown in Table 4, the AlpacaEval turn-taking success rate indicates that this phenomenon can be observed across many models. We suppose that short questions require the model to switch between listening state and speaking state quite fast, which is hard to learn.
>
> 2) In real-world testing, it is best to maintain a quiet environment, as our model was not trained with any data augmentation techniques involving additive noise. When the environment is noisy, the model’s responses are often unreasonable. In future work, we plan to further explore strategies to enhance the model’s robustness in noisy conditions.
>
> 3) Regarding the sentences used for contextual-independent barge-in, for the sake of implementation simplicity, we limited the training set to only 10 interruption phrases. While the model demonstrated a certain degree of generalization during testing, some sentences still failed to successfully interrupt the model—particularly when the interruption was very short or spoken at a low volume. In such cases, the model may confuse the input with backchanneling.
>
> - **Q5: How would the model scale to multilingual or code-switched dialogue?**
>
> This is indeed an interesting and worthwhile direction to explore. However, we acknowledge that at the current stage, SALMONN-Omni focuses solely on English speech input and output due to the data constraint. We believe it's valuable to extend SALMONN-omni to handle multilingual situations by adding multilingual or code-switched data. However, it will need careful exploration of how to balance multiple languages.
>
> - **Real-time deployment user-machine interaction experiments**
>
> We deployed our model in a real-world environment and assessed its performance through live human-model interactions across three primary scenarios: multi-turn dialogue, context-independent barge-in, and context-dependent barge-in. For each scenario, we collected 20 dialogue sessions due to time constrain. In the multi-turn dialogue scenario, conversations consisted of an average of 5 turns, with a mean duration of 75 seconds. In both barge-in scenarios, the dialogues had an average length of 3 turns.
>
> 1) *Turn-taking statistics and latency analysis*
>
> |               | IPU     | Pause   | Gap     |
> |---------------|---------|---------|---------|
> | Ours          | 61.89s  | 5.49s   | 6.50s   |
> | Ground Truth  | 63.3s   | 4.68s   | 5.01s   |
>
> The results here (normalized to 5-turn dialogues) illustrate the turn-taking statistics of SALMONN-omni compared with our multi-turn training data (noted as Ground Truth). Furthermore, our model demonstrates an average response latency of **405 ms**, largely attributed to its current architecture, which necessitates a wait for four time blocks to accumulate adequate context for the speech synthesizer. We anticipate that this latency can be further minimized through targeted optimization of the synthesizer. In the context-independent and context-dependent barge-in scenarios, the response latency upon interruption is approximately **128 ms** and **144 ms**, respectively, suggesting that the model is capable of responding promptly to user interruptions.
>
> 2) *Quality assessment of the generated content*
>
> Although time constraints prevented us from conducting a rigorous human evaluation, we adopted the reviewer’s suggestion and assessed the quality of the model’s responses in real-world conversations using ChatGPT Score from three perspectives: naturalness, fluency, and overall quality.
>
> **Naturalness** evaluates how human-like, smooth, and contextually appropriate the responses are; while **Fluency** evaluates the grammatical correctness, flow, and ease of understanding of the language used. All aspects were rated on a scale from 1 to 5, with higher scores indicating better performance.
>
> |               | naturalness | fluency | overall |
> |---------------|-------------|---------|---------|
> | Ours          | 3.80        | 4.00     | 3.65    |
> | Ground Truth  | 4.14        | 4.62    | 4.18    |
>
> The results indicate that our model is still able to produce reasonable responses during real-world testing. However, a slight gap in fluency compared to the ground truth is observed, which can be attributed to inevitable errors introduced during ASR, as well as the responses generated by our model tend to be more conversational and may not strictly adhere to formal grammatical rules.

---

> > ### Comment · Area_Chair_jwMV · 2025-08-06
> > **Please respond to authors' rebuttal!**
> >
> > Dear reviewer 2yL6,
> > Please check the authors' rebuttal, engage in a discussion if there are remaining concerns or write them a note/consider changing your review and/or ratings, if you do not have any additional concerns.
> > Thanks,
> > AC

---

### Official Review · Reviewer_ExHa · 2025-07-03

**Clarity:** 3
**Significance:** 2
**Originality:** 2
**Rating:** 4
**Confidence:** 4

**Summary:**

This paper introduces SALMONN-omni, a standalone, full-duplex speech Large Language Model that innovatively operates without injecting audio codecs into its vocabulary. It achieves the ability to listen and speak simultaneously by seamlessly integrating a streaming speech encoder, a unified LLM backbone, and a streaming speech synthesizer. The model adopts a novel "explicit thinking" strategy, training the LLM to generate special control tokens to autonomously decide when to start speaking and when to remain listening, thereby managing conversational state transitions. Experimental results show that SALMONN-omni excels at handling complex conversational dynamics such as turn-taking, backchanneling, and context-dependent barge-ins, with its performance significantly surpassing existing open-source full-duplex models.

**Questions:**

- The model generates a large number of `<think>` tokens due to the mismatch in sequence length between text and speech, particularly when it has finished generating a text response but is still in the "speaking" state. A key question is whether this process of generating numerous non-substantive tokens could affect the model's core text-based abilities, especially in the multi-turn conversation scenarios.

**Ethical Concerns:**

["NO or VERY MINOR ethics concerns only"]

**Final Justification:**

The authors report multi-turn and barge-in evaluations in the rebuttal, which addresses the original review’s core concerns.

**Limitations:**

yes

**Quality:**

3

**Strengths And Weaknesses:**

## Pros

- The proposed model handles both speech understanding and generation outside of the LLM, which largely preserves the LLM's inherent capabilities. Results on several spoken QA benchmarks show that it outperforms many existing models, including strong models like Kimi-Audio and Qwen2.5-Omni.

- The paper is well-written and easy to understand.


## Cons

- The model needs to synthesize its output tokens into speech and then re-encode that speech through its own encoder as an "assistant stream" input. Compared to models like Moshi, which only need to perform auto-regression on codec tokens , this synthesis and re-encoding loop adds inference overhead and impacts latency.

- For the full-duplex evaluation, the authors focused on metrics like success rate and F1 scores but did not report measurements of actual inference latency. A more thorough evaluation should assess the actual time of the gap or overlap in different full-duplex scenarios (turn-taking, interruption, backchanneling).

- Certain design choices, such as the use of a Mamba-based speech encoder, are not accompanied by sufficient experimental analysis to justify their necessity—for example, by detailing its impact on response quality and inference speed.

- The current evaluation is limited to single-turn interactions. I believe that an evaluation in multi-turn scenarios is also necessary for a more comprehensive assessment.

---

> ### Author Rebuttal · Authors · 2025-07-31
>
> Thank you for your efforts—your suggestions are extremely valuable for improving our paper. We hope that the additional experiments we have provided will help address your concerns.
>
> - **Q1: The model needs to synthesize its output tokens into speech and then re-encode that speech through its own encoder as an "assistant stream" input. Compared to models like Moshi, which only need to perform auto-regression on codec tokens , this synthesis and re-encoding loop adds inference overhead and impacts latency.**
>
> Thank you for the insightful comment. We would like to clarify that the Environment Stream and Assistant Stream are processed in parallel which does not necessarily increase the latency. Furthermore, during real-world deployment, our approach does not require the Environment Stream and Assistant Stream to be fully aligned, and thus does not incur additional latency. While a potential concern is the overhead introduced by interleaving all embeddings into a single sequence, we note that this design choice allows the LLM backbone to model the flexible interaction between the two streams. In contrast, Moshi adopts a RQ-Transformer component, which compresses information into a more compact representation. Our interleaving approach trades off some efficiency for improved alignment and interaction modeling, which we found beneficial in our full-duplex dialogue setting. In real-world deployment, the latency introduced by this design is also acceptable. Processing one time block (80 ms) of input with the encoder takes approximately 55 ms, while the LLM requires around 70 ms to decode and generate the response for one time block.
>
> - **Q2: For the full-duplex evaluation, the authors focused on metrics like success rate and F1 scores but did not report measurements of actual inference latency. A more thorough evaluation should assess the actual time of the gap or overlap in different full-duplex scenarios (turn-taking, interruption, backchanneling).**
>
> Thank you for your suggestion. Here, we conducted a detailed latency evaluation and analyzed the turn-taking statistics of the generated dialogues. We deployed our model in a real-world setting and conducted human-model interactions through live conversations to evaluate its performance across three main scenarios: multi-turn dialogue, contextual-independent barge-in and contextual-dependent barge-in. We collected 20 dialogue sessions for each scenario. In the multi-turn dialogue setting, conversations averaged 5 turns with a mean duration of 75 seconds. In the barge-in scenarios, the dialogues averaged 3 turns.
>
> |               | IPU     | Pause   | Gap     |
> |---------------|---------|---------|---------|
> | Ours          | 61.89s  | 5.49s   | 6.50s   |
> | Ground Truth  | 63.3s   | 4.68s   | 5.01s   |
>
> The turn-taking statistics in a multi-turn dialogue scenario are shown in the table (normalized to 5-turn dialogues), and our model exhibits an average response latency of **405 ms**, primarily due to the current design, which requires waiting for four time blocks to provide sufficient context for the speech synthesizer. We believe this latency can be further reduced by optimizing the synthesizer. In contextual-independent and contextual-dependent barge-in scenarios, the response latency when interrupted is approximately **128 ms** and **144 ms**, respectively, indicating that our model can quickly react to user interruptions.
>
> - **Q3: Certain design choices, such as the use of a Mamba-based speech encoder, are not accompanied by sufficient experimental analysis to justify their necessity—for example, by detailing its impact on response quality and inference speed.**
>
> This represents an alternative design choice rather than the central focus of our work. Any architecture capable of handling streaming speech processing—such as CNNs or causal transformers—can certainly be employed. We opt for Mamba due to its ability to perform streaming sequence modeling with comparatively lower computational overhead than causal transformers, while also demonstrating satisfactory performance in our preliminary evaluations.
>
> - **Q4: The current evaluation is limited to single-turn interactions. I believe that an evaluation in multi-turn scenarios is also necessary for a more comprehensive assessment.**
>
> We evaluated the model’s multi-turn dialogue capabilities in both simulated and real-world scenarios. In simulated settings, we follow the evaluation protocol of SLAM-omni, assessing the model’s final-turn output using the ChatGPT Score  [1]. The evaluation covered dialogues ranging from 2 to 5 turns. Test data were collected using the same method employed for constructing the multi-turn dialogue training data, with 50 samples gathered for each dialogue length, resulting in a total of 200 test samples.
>
> |   N Turn     |  2        | 3        | 4        | 5        |
> |----------------|--------------|--------------|--------------|--------------|
> | GLM-4-Voice     | 4.02 (3.52)  | 4.66 (3.98)  | 4.52 (3.84)  | 4.36 (3.62)  |
> | MiniCPM-o       | 4.38 (4.20)   | 4.40 (4.16)   | 4.20 (3.96)   | 4.44 (4.16)  |
> | Qwen2.5-omni    | 4.28 (4.02)  | 4.44 (4.20)   | 4.30 (4.08)   | 4.10 (3.92)   |
> | Freeze-Omni     | 3.58 (2.94)        | 4.00 (3.12)         | 3.44 (2.58)        | 3.34 (2.46)        |
> | ours            | 4.24 (4.04)  | 4.48 (4.18)  | 4.44 (4.16)  | 4.34 (4.02)  |
>
> We compared our model with several full-duplex and half-duplex models. The results (numbers in parentheses are S2S performance) show that our model maintains stable performance across dialogues of varying lengths. Moreover, it significantly outperforms the open-source full-duplex model Freeze-Omni and achieves performance comparable to other half-duplex models, consistent with the findings from other evaluations presented in our paper.
>
> As for real-world settings, we transcribed the dialogues into text and evaluated aspects such as naturalness and fluency using the ChatGPT Score. **Naturalness** evaluates how human-like, smooth, and contextually appropriate the responses are; while **Fluency** evaluates the grammatical correctness, flow, and ease of understanding of the language used. All aspects were rated on a scale from 1 to 5, with higher scores indicating better performance. Comparisons with the ground truth further demonstrate the multi-turn dialogue capability of our model in real-world conversational settings. A slight discrepancy in fluency compared to the ground truth can be attributed to unavoidable errors introduced by the ASR system, as well as the tendency of our model’s responses to be more conversational in nature and less constrained by formal grammatical conventions.
>
> |               | naturalness | fluency | overall |
> |---------------|-------------|---------|---------|
> | Ours          | 3.80        | 4.00     | 3.65    |
> | Ground Truth  | 4.14        | 4.62    | 4.18    |
>
> [1] Wenxi Chen, Ziyang Ma, Ruiqi Yan, et al. “SLAM-Omni: Timbre-Controllable Voice Interaction System with Single-Stage Training”. In Proc. ACL, Vienna, 2025.
>
> - **Q5: The model generates a large number of \<think\> tokens due to the mismatch in sequence length between text and speech, particularly when it has finished generating a text response but is still in the "speaking" state. A key question is whether this process of generating numerous non-substantive tokens could affect the model's core text-based abilities, especially in the multi-turn conversation scenarios.**
>
> Thank you for your question. All speech dialogue models with streaming generation capabilities must address the challenge of sequence length mismatch. A common solution is to pad the text sequence to match the length of the speech sequence; accordingly, models such as Moshi generate padding tokens in the text stream output. We further explored various options for this "thinking content" (the padding portion of the text sequence), including simple pad tokens, question ASR transcriptions, and random text, as detailed in Appendix C. Our results show that generating a dedicated \<think\> token as padding yields the best performance. We hypothesize that using a special token (e.g., \<think\>) when the model is not producing a meaningful answer makes state transitions more distinct and easier for the model to learn. We also compared our model’s performance, using both text ("Ours-text") and audio inputs ("Ours-wav"), to the original LLaMA-3. The performance degradation for the LLM was minimal. Furthermore, multi-turn dialogue evaluations confirm that this design does not suffer from noticeable performance loss as the number of dialogue turns increases.
>
> ||Llama Q.|Web Q.|Trivia Q.|AlpacaEval|
> |---|---|---|---|---|
> |Llama-3|0.85|0.56|0.73|3.83|
> |Ours-text|0.84|0.52|0.68|3.82|
> |Ours-wav|0.80|0.51|0.66|4.05|

---

> > ### Comment · Reviewer_ExHa · 2025-08-05
> >
> > Thank you for your response. My concerns have been largely addressed, and I will raise my score.

---

> > > ### Author Response · Authors · 2025-08-06
> > > **further response**
> > >
> > > Once again, we sincerely appreciate your valuable suggestions for our work, as well as your recognition of our paper.

---

### Official Review · Reviewer_sYnD · 2025-07-03

**Clarity:** 2
**Significance:** 4
**Originality:** 3
**Rating:** 4
**Confidence:** 4

**Summary:**

This paper introduces SALMONN-omni, a system designed to enable full-duplex spoken dialogues without relying on neural codecs, unlike several prior approaches. The method introduces a “thinking” mechanism to alternate between speaking and listening modes within a single sequence; however, the explanation of this mechanism is vague, and the core idea remains difficult to grasp. Additionally, it leverages DPO to improve naturalness in barge-ins and backchanneling, resulting in a three-stage training pipeline. While the model demonstrates state-of-the-art results on several public benchmarks compared to existing open-weight systems, the comparisons are not entirely straightforward. The training data, architecture, and training methodology differ significantly from other models, making it unclear which factors are responsible for the performance gains.

**Questions:**

- Unclear justification for avoiding neural codecs: The paper does not clearly explain the limitations of neural codecs or provide experimental validation for avoiding them. Are there any ablation studies or comparisons demonstrating the effectiveness of the proposed method without a neural codec?
- Sound quality evaluation is lacking: One of the main advantages of neural codecs is their ability to generate high-quality audio. Therefore, the proposed method without neural codes should include sufficient discussion or analysis of the sound quality produced by the proposed method. Are there any objective or subjective evaluations to support its adequacy? A comparison would strengthen this point.
- Ambiguity in the "thinking" mechanism: The core concept of the proposed “thinking” strategy is difficult to understand. Notations such as <bos> and <and> are introduced without adequate explanation. More detailed clarification, especially in relation to Figure 2, is necessary to make this part accessible to readers.
- Reproducibility concerns: Although the paper appears to rely mostly on public datasets, it also uses TTS-generated spoken text. If the TTS resources or generated data are not publicly available, full reproduction of the experiments may be challenging. Please clarify the availability of these resources.
- Lack of structured dataset summary: Section 4.3 could be improved by presenting the dataset information in a structured format, such as a table. This would help readers understand the experimental setting more easily.
- Insufficient baseline details: The paper lacks important details regarding the setup of each baseline method. Without this, it is difficult to assess the fairness and significance of the comparisons. Please provide more comprehensive descriptions to support the experimental findings.

**Ethical Concerns:**

["NO or VERY MINOR ethics concerns only"]

**Final Justification:**

I still have concerns regarding the clarity of the “thinking” strategy, which I believe requires major revision, and I cannot fully assess it without carefully reviewing the revised version. However, regarding my other concerns about the complex training strategies, the authors’ point that similar complexity exists in other approaches has changed my perspective. Therefore, I have adjusted my recommendation from 3: Borderline Reject to 4: Borderline Accept.

**Limitations:**

While the paper touches on certain technical limitations, it lacks discussion of the broader societal impacts. For instance, techniques of this nature often raise concerns about hallucinations, yet this issue is not addressed. Additionally, potential biases—such as skewed conversational topics or limited speaker demographic coverage—should be considered and discussed more explicitly.

**Paper Formatting Concerns:**

No concern

**Quality:**

3

**Strengths And Weaknesses:**

Strengths
- Proposes a full-duplex dialogue system that avoids the use of neural codecs, enabling a simpler and unified LLM-based modeling approach.
- Demonstrates state-of-the-art performance on several benchmarks.

Weaknesses
- The explanation of the core “thinking” strategy is vague and difficult to follow, making the overall method hard to understand.
- The training pipeline is complex, involving a three-stage process including DPO which may hinder reproducibility and practical adoption.

---

> ### Author Rebuttal · Authors · 2025-07-31
>
> We sincerely thank the reviewer for the effort. We hope the following responses will help address your concerns.
> - **Q1: The explanation of the core “thinking” strategy is vague ...**
>
> Thank you for pointing out these issues, and we will improve the presentation in the final version. Here, we would like to clarify the motivation behind our "thinking" strategy, which is designed to handle the dynamics of natural conversations in full-duplex speech dialogue systems.
>
> Full-duplex models alternate between two states: listening, where the model remains silent but attentive, and speaking. In the speaking state, the LLM generates speech while continuing to listen, which is required to manage natural conversation behaviours such as barge-ins. In the listening state, the model still needs to perform internal computation to monitor ambient sounds and detect potential incoming speech, even though it does not produce spoken output. We refer to this process as thinking, drawing an analogy to human behaviour.
>
> In SALMONN-omni, we explored different thinking strategies, primarily distinguishing them based on whether the model feeds the thinking content back into the input, and present the results in Table 2. Implicit thinking strategy requires the model to make decisions between the \<listen\> and \<speak\> state tokens in each time block and these tokens are not fed back into the model’s input. However, the model trained with explicit thinking strategy feeds all the generated tokens (both \<think\> and its response tokens) into its input and uses \<shift\> to mark the state switching between listening and speaking. Experiments show that explicit thinking consistently outperforms implicit thinking, likely due to better alignment with the autoregressive nature of LLMs.
> We also experimented with different forms of thinking content, as detailed in Appendix C. Among them, the single \<think\> token outperformed alternatives such as ASR transcriptions and random text, suggesting it provides the most effective signal for this internal state.
>
> Lastly, we were unsure what the notation "[object Object]" in your comment was intended to reference. We would greatly appreciate it if the reviewer could clarify this point.
>
> - **Q2: The training pipeline is complex, ...**
>
> We will open-source our training data and source code to facilitate reproducibility. Our three-step training procedure is straightforward and closely follows the standard paradigm of (multimodal) LLM training. Each stage is essential for achieving high-performance speech dialogue interaction: 1) the first stage focuses on speech understanding; 2) the second on speech generation; 3) the final DPO stage enhances full-duplex capabilities to enable more natural conversational experiences.
>
> - **Q3: Unclear justification for avoiding neural codecs ...**
>
> Injecting codecs into the LLM is certainly an alternative approach. We are not drawing any definitive conclusions about which paradigm future duplex models must adopt and merely exploring a framework that has not been investigated in prior work. However, major challenges in integrating autoregressive LLMs with neural codecs include the following:
>
> 1) Mismatch in unit durations: Injecting speech codecs often degrades the language processing performance of LLMs [1].  This is believed to be largely attributed to the mismatch between the variable-length word pieces used by LLMs and the short, fixed-duration units typically produced by speech codecs.
>
> 2) Content inconsistency: In neural codec-based approaches [2], speech tokens are often generated sequentially after text tokens. As a result, the content of the speech response can noticeably differ from its corresponding text response. This issue (commonly referred to as the "modality gap" problem), leads users to perceive that the spoken output lacks the intelligence or coherence of the text response.
>
> For our approach, which does not require injecting the codec into the LLM, the above two issues are significantly less problematic. As shown in the following table, we compared the performance of our model using both text and audio inputs with the original performance of LLaMA-3. It turns out that the performance degradation of the LLM is not as significant as Moshi and our model exhibits comparable performance when using text and audio inputs.
>
> ||Llama Q.|Web Q.|Trivia Q.|AlpacaEval|
> |---|---|---|---|---|
> |Llama-3|0.85|0.56|0.73|3.83|
> |Ours-text|0.84|0.52|0.68|3.82|
> |Ours-wav|0.80|0.51|0.66|4.05|
>
> Moreover, we found that the "modality gap" problem becomes severer for codec-based models in multi-turn dialogue scenarios. We followed the multi-turn dialogue setup used in SLAM-Omni to evaluate the multi-turn conversational capabilities of different models. The results (numbers in parentheses indicate S2S results) show that while GLM-4-Voice demonstrates stable performance in S2T, its S2S performance is significantly inferior. We observed that the generated audio responses often begin with phrases like "You want a friend? You got one…", which are not present in the corresponding textual responses.
>
> |N Turn|2|3|4|5|
> |---|---|---|---|---|
> |GLM-4-Voice|4.02 (3.52)|4.66 (3.98)|4.52 (3.84)|4.36 (3.62)|
> |Ours|4.24 (4.04)|4.48 (4.18)|4.44 (4.16)|4.34 (4.02)|
>
> [1] Jiatong Shi, Chunlei Zhang, Jinchuan Tian, et al. “Balancing Speech Understanding and Generation Using Continual Pre-training for Codec-based Speech LLM”. In: arXiv preprint arXiv:2502.16897 (2025).
>
> [2] Aohan Zeng, Zhengxiao Du, Mingdao Liu, et al. “GLM-4-Voice: Towards intelligent and human-like end-to-end spoken chatbot”. In: arXiv preprint arXiv:2412.02612 (2024).
>
> - **Q4: Sound quality evaluation is lacking ...**
>
> We compared the UTMOS scores of different models across various datasets, and the results indicate that the audio generated by our model is of high quality.
>
> |                  | Llama Q.| Web Q. | Trivia Q. | AlpacaEval |
> |------------------|-------|-------|--------|--------|
> | **Predicted turn-taking** |       |       |        |        |
> | moshi            | 3.245 | 3.257 | 3.017  | 3.193  |
> | freeze-omni      | 4.288 | 4.273 | 4.293  | 4.255  |
> | SALMONN-omni     | 4.256 | 4.250 | 4.272  | 4.243  |
> | **Oracle turn-taking**    |       |       |        |        |
> | GLM-4-Voice      | 4.180 | 4.107 | 4.145  | 3.890  |
> | Qwen2.5-Omni     | 4.140 | 4.124 | 4.127  | 3.950  |
> | VITA-1.5         | 4.042 | 4.091 | 4.104  | 3.852  |
> | miniCPM-o        | 4.171 | 4.152 | 4.155  | 4.110  |
> | Kimi-Audio       | 2.882 | 2.713 | 2.793  | 2.327  |
> | Baichuan-Audio   | 4.261 | 4.216 | 4.292  | 3.959  |
> | Freeze-Omni      | 4.300 | 4.281 | 4.278  | 4.245  |
> | SALMONN-omni     | 4.262 | 4.263 | 4.279  | 4.240  |
>
> - **Q5: Reproducibility concerns ...**
>
> We promise to open-source all training data and source code used in our paper to facilitate reproducibility. For our self-generated dialogue data, all models involved in the pipeline are publicly available. The questions and topics are selected from existing text datasets such as WebQuestions and Natural Questions. Text responses or dialogues are generated using Llama-3-8B-Instruct, while both the questions and corresponding speech audio are synthesized using CosyVoice2-0.5B. Both models are open-source.
>
> - **Q6: Lack of structured dataset summary ...**
>
> Thanks for your advice. Due to the page limit, we put the dataset details in Appendix A, shown in Table 7. We will consider rearranging our presentation to enable better understanding.
>
> - **Q7: Insufficient baseline details ...**
>
> We will add more details about baselines in the Appendix in the final version. In our paper, all baseline models are opensourced models and we have correctly referenced all baseline papers. For all baseline models, we test them following the official recipes with preferenced prompts and default decoding options to ensure all models demonstrate the best performance of their own. And we believe it can support a relatively fair comparison.
>
> - **Q8: Limitations**
>
> We thank the reviewer for highlighting these important limitations related to societal impact, and we will incorporate them into the limitations section of the paper. These are fundamental non-technical challenges shared by all conversational AI systems and will require sustained, long-term efforts from both the research community and society at large to address.

---

> > ### Comment · Reviewer_sYnD · 2025-08-05
> >
> > Thank you for the detailed explanations. However, I still find that the core “thinking” strategy remains insufficiently explained. While the response provided some clarity, making this aspect truly understandable would require significant additional effort. Furthermore, the complexity of the training pipeline remains a concern. For these reasons, I would prefer to keep my score as it is.

---

> > > ### Author Response · Authors · 2025-08-06
> > > **further response**
> > >
> > > Thank you for your valuable feedback, which will help us further improve our manuscript.
> > >
> > > 1. To address your concern about the “thinking” mechanism, we plan to include a detailed illustration in the appendix, visually demonstrating how a sample is trained under this mechanism. We believe this will clarify our algorithm’s process and improve interpretability.
> > >
> > > 2. Regarding the three-stage training method, we understand your concerns about complexity. However, our approach is relatively streamlined compared to other spoken dialogue LLMs. For instance, Moshi also adopts a three-stage process (pre-training, post-training, and fine-tuning) while in Qwen2.5-Omni, both the Thinker and Talker modules each require three separate training phases.
> > >
> > > 3. The third stage in SALMONN-omni employs Direct Preference Optimization (DPO) to enhance the model’s ability to manage conversational dynamics, which is a key contribution of our work. DPO is a standard technique in both text- and image-language LLM training. If incorporated into other full-duplex spoken dialogue LLMs (e.g., Moshi, Freeze-Omni), it would similarly add an additional stage to their pipelines.
> > >
> > > Overall, we humbly believe that your concerns regarding the clarity and complexity of our method can be addressed to a certain extent through further discussion. Therefore, if you have any specific concerns, we would greatly welcome your feedback.

---

### Note · Authors · 2025-08-15

Dear AC,

We sincerely thank the Area Chair and all reviewers for their time, effort, and constructive feedback during the review and rebuttal process.

We appreciate the recognition of our paper’s novelties and strengths, including: its well-motivated and clearly written presentation (ExHa, 2yL6), the novel yet simple duplex modeling architecture (sYnD, 2yL6), RL-based duplex interaction optimization (2yL6, tsTQ), the comprehensive evaluation (2yL6, tsTQ), and the strong performance on QA and dialogue tasks (sYnD, ExHa, 2yL6).

We have addressed all reviewer concerns in detail during rebuttal and discussion. During the discussion stage, three reviewers participated. After we supplemented our work with assessments of speech quality, multi-turn dialogue evaluation, and tests of the model’s performance and latency in real-world conversations, two of them (ExHa, tsTQ) explicitly stated that their concerns were largely resolved and indicated they would raise their scores. Reviewer sYnD maintained concerns regarding the “thinking” strategy and the training pipeline’s complexity, but did not re-engage after our clarifications. Given the positive and specific acknowledgments of these elements from other reviewers — e.g.,
- "The model adopts a novel 'explicit thinking' strategy" in Summary of Reviewer ExHa
- "The paper is well-written and easy to understand." in Strengths of Reviewer ExHa
- "Well-motivated design: The 'thinking' strategy and stream interleaving offer a clean integration of speech and dialogue dynamics." in Strengths of Reviewer 2yL6
- "A well-executed study: Efficiently enables end-to-end speech via encoder/synthesizer integration while freezing the LLM backbone. The explicit state strategy reduces duplex system complexity" in Strengths of Reviewer tsTQ

and the novelty and effectiveness of involving DPO in training
- "Application of DPO to barge-in and turn-taking tasks is novel and well-justified." in Strengths of Reviewer 2yL6
- "DPO further boosts performance." in Strengths of Reviewer tsTQ,

we humbly believe that our responses sufficiently address Reviewer sYnD's remaining concern.

We will integrate all reviewer suggestions into the main text and appendix to ensure the camera-ready version meets the highest standards of clarity and completeness. We greatly appreciate the reviewers' and the area chair's dedication to a thorough and constructive review process, which has significantly contributed to improving the quality of our paper.

---

### Decision · Program_Chairs · 2025-09-17

**Decision:**

Accept (poster)

**Comment:**

This paper proposes SALMONN-omni, a standalone full-duplex speech LLM that eliminates audio codecs in token space (as was used by Moshi and other earlier systems) and incorporates a dynamic thinking mechanism for seamless turn-taking within the LLM backbone. It outperforms existing open-source full-duplex models by at least 30%, performs competitively to half-duplex and turn-based systems despite using less training data, and shows strong performance in complex conversational scenarios like barge-in, echo cancellation, and backchanneling.

Summary of Strengths:
- The work introduces a novel, clean and unified architecture, with a well-motivated design including a novel “thinking” mechanism and RL-based training (DPO) for modeling turn-taking and barge-in dynamics.
- The experiments demonstrates that the approach is also effective and outperforms strong baselines, despite using less data.
- The paper is mostly clearly written and easy to follow, though there are some parts that are less clear (please see the weaknesses).

Summary of Weaknesses:
- The core “thinking” strategy can be better explained. There were useful discussions with the reviewers during the rebuttals, which I believe will help with clarifying these parts of the paper.
-  The three-stage training pipeline (including DPO) adds complexity. During the rebuttals authors suggested that they will release their code, which will help with reproducability.
- Reviewers highlighted several issues with the evaluation, such as lack of evaluation on multi-turn datasets. Authors included additional results to address these concerns during the rebuttals.

The discussions during the rebuttals resulted in three reviewers accepting the author responses and increases in their scores. Once these  clarifications and additional results are integrated into the paper, it will be an interesting contribution.